# Tiny but Mighty: Small RNAs—The Micromanagers of Bacterial Survival, Virulence, and Host–Pathogen Interactions

**DOI:** 10.3390/ncrna11030036

**Published:** 2025-05-05

**Authors:** Rajdeep Banerjee

**Affiliations:** Department of Biomolecular Chemistry, University of Wisconsin-Madison, Madison, WI 53706, USA; rbanerjee8@wisc.edu

**Keywords:** small RNA, bacterial pathogenesis, stress response, antibiotic resistance, quorum sensing, host–pathogen interactions

## Abstract

Bacterial pathogens have evolved diverse strategies to infect hosts, evade immune responses, and establish successful infections. While the role of transcription factors in bacterial virulence is well documented, emerging evidence highlights the significant contribution of small regulatory RNAs (sRNAs) in bacterial pathogenesis. These sRNAs function as posttranscriptional regulators that fine-tune gene expression, enabling bacteria to adapt rapidly to challenging environments. This review explores the multifaceted roles of bacterial sRNAs in host–pathogen interactions. Firstly, it examines how sRNAs regulate pathogenicity by modulating the expression of key virulence factors, including fimbriae, toxins, and secretion systems, followed by discussing the role of sRNAs in bacterial stress response mechanisms that counteract host immune defenses, such as oxidative and envelope stress. Additionally, this review investigates the involvement of sRNAs in antibiotic resistance by regulating efflux pumps, biofilm formation, and membrane modifications, which contribute to multi-drug resistance phenotypes. Lastly, this review highlights how sRNAs contribute to intra- and interspecies communication through quorum sensing, thereby coordinating bacterial behavior in response to environmental cues. Understanding these regulatory networks governed by sRNAs is essential for the development of innovative antimicrobial strategies. This review highlights the growing significance of sRNAs in bacterial pathogenicity and explores their potential as therapeutic targets for the treatment of bacterial infections.

## 1. Introduction

The human body harbors a complex and diverse microbiota that colonizes nearly all organs, including the gastrointestinal tract, skin, and respiratory system. Bacterial cells are estimated to be present in numbers roughly equal to human cells [1]. While many of these microorganisms contribute to host health through mutualistic relationships, humans are also frequently exposed to pathogenic bacteria capable of establishing infections. The human host employs multiple strategies to curb bacterial infections, including physical barriers, immune responses, and antimicrobial molecules [2]. For example, the skin and mucosal surfaces serve as the first line of defense, preventing bacterial entry through tight junctions and mucus secretion, which entraps pathogens such as *Pseudomonas aeruginosa* [3]. Innate immune responses further combat infections through the production of antimicrobial peptides (AMPs) like defensins and cathelicidins, which disrupt bacterial membranes, effectively targeting pathogens such as *Staphylococcus aureus* [4,5] and *Mycobacterium tuberculosis* (Mtb) and also aid protection against viruses including Human Immunodeficiency Virus (HIV) [6]. In addition, the complement system enhances bacterial clearance by opsonization, and membrane attack complex formation, particularly against *Neisseria meningitidis* [7]. Furthermore, the host cells limit bacterial growth by sequestering essential nutrients, a strategy known as nutritional immunity; for instance, lactoferrin and calprotectin bind iron and zinc to restrict the proliferation of uropathogenic strains of *Escherichia coli* and *Klebsiella pneumoniae* in the urinary tract [8,9,10]. Adaptive immunity further strengthens bacterial control through antibody production and T-cell-mediated responses, which are critical for long-term protection against Mtb [11]. These multilayered defense mechanisms collectively work to prevent bacterial infections and maintain host homeostasis. To circumvent these host defenses and ensure their survival inside the host and establish an infection, the bacterial pathogens have evolved sophisticated virulence factors and immune evasion strategies [12,13,14]. These include spatiotemporal expression of virulence factors as well as stress-responsive pathways that equip the pathogen with the necessary adaptations needed to colonize and persist within the host. In addition, bacterial pathogens often communicate with each other through chemical messengers, which are collectively called quorum sensing [15,16,17,18]. Bacterial quorum sensing regulates group behaviors such as biofilm formation, virulence factor expression, and antibiotic resistance. By sensing and responding to host-derived signals, bacteria can coordinate adaptive responses that enhance colonization, persistence, and immune evasion. Finally, the bacteria have evolved a series of strategies to curb the effect of antibiotics including modification of the lipid bilayer, expression of efflux pumps, and the inhibition of antibiotic uptake systems [19,20,21]. Together, the bacteria have developed a sophisticated system to evade host immunity, and studying the molecular mechanisms behind these processes could lead to new therapeutic advancements. Although the role of various proteins in virulence of these pathogens have been well characterized, the importance of bacterial small regulatory RNA was only recognized in the last decade or so.

Bacterial sRNAs are a diverse class of molecules, typically ranging from 40 to 500 nucleotides, that play essential roles in posttranscriptional gene regulation by interacting with target mRNAs or proteins. Initially categorized as small non-coding RNAs, only a handful of these sRNAs such as SgrS from enteric species, SR1 from *Bacillus subtilis*, Pel RNA from *Streptococcus pyogenes*, RNAIII, and Psm-Mec from *Staphylococcus aureus* have been found to encode small peptides [22]. sRNAs are involved in key cellular processes, including the regulation of bacteria transport through the modulation of outer membrane protein expression, iron homeostasis, quorum sensing, and bacterial virulence [23,24,25,26,27]. Advancements in bioinformatics and experimental methodologies have facilitated the discovery of over 900 bacterial sRNAs, primarily transcribed from intergenic regions [28]. These sRNAs are classified as either cis-encoded or trans-encoded based on their genomic arrangement relative to target genes. Cis-encoded sRNAs are transcribed from the same genomic locus as their target mRNA and typically exhibit perfect complementarity. In contrast, trans-encoded sRNAs are transcribed from separate loci and regulate multiple targets with partial complementarity, often requiring RNA chaperones like Hfq [29,30]. Computational approaches, including comparative genomics and machine learning-based predictions, have significantly improved sRNA identification and target prediction, although experimental validation remains essential due to challenges posed by imperfect base pairing in trans-encoded sRNAs. The continued discovery and functional characterization of sRNAs are critical to understanding their roles in bacterial stress response and pathogenesis [31].

Bacteria encounter numerous stressors within the host environment, including nutrient deprivation, temperature fluctuations, oxidative stress, osmotic pressure changes, and exposure to antimicrobial compounds [32,33]. To mitigate the harmful effects of the aforementioned host response, bacteria have developed intricate stress response pathways regulated through transcriptional control by sigma factors and two-component systems [34,35,36]. The discovery of sRNAs has unveiled an additional layer of posttranscriptional regulation, enabling bacteria to fine-tune gene expression with high precision and efficiency. These small RNAs regulate the expression of their target gene by altering mRNA stability, translation, and processing, allowing rapid adaptation to environmental changes. By coordinating cellular processes such as metabolic reprogramming [37,38], detoxification [39], DNA repair [40], and biofilm formation [41,42], sRNAs serve as pivotal regulators of bacterial stress responses. Beyond stress adaptation, sRNAs also play a central role in bacterial virulence, governing the expression of key pathogenicity determinants, including toxins [43], adhesins, invasins [44], and immune evasion factors [45]. Furthermore, sRNAs have been shown to play a key role in antimicrobial resistance, one of the pressing global concerns.

The rise in antimicrobial resistance (AMR) where the pathogenic bacteria become resistant to multiple drugs represents a significant global health threat, with resistant infections leading to increased morbidity, mortality, and healthcare costs [46]. The World Health Organization (WHO) has identified 24 priority pathogens exhibiting significant resistance, including AMR Mtb and Gram-negative bacteria resistant to last-line antibiotics [47]. Projections indicate that AMR could result in approximately 39 million deaths between 2025 and 2050, underscoring the urgency of addressing this crisis.

While mechanisms such as horizontal gene transfer and mutations in antibiotic target sites have been extensively studied [48,49], recent research has identified sRNAs as important contributors to antimicrobial resistance [50,51]. These sRNAs influence antibiotic resistance through multiple pathways including the modification of cell membrane components, the repression of antibiotic uptake systems, the activation of efflux pumps, and the generation of biofilms [51]. In addition, bacteria can also achieve antimicrobial resistance through an alternative mechanism—quorum sensing [52]. This process enables bacteria to respond to specific chemical messengers by fine-tuning the expression of genes involved in biofilm formation and antimicrobial resistance, allowing them to adapt to changing environmental conditions [52]. This dual functionality highlights the versatility of small RNAs in shaping bacterial survival and community dynamics.

Bacteria exist within complex microbial communities, where interspecies communication is essential for survival with the host. Quorum sensing is a bacterial communication process that enables populations to coordinate gene expression based on cell density. This signaling mechanism relies on the production, release, and detection of small signaling molecules called autoinducers. As bacterial populations grow, the concentration of these molecules increases, allowing bacteria to sense their community size and regulate various behaviors collectively. Quorum sensing controls a wide range of functions, including biofilm formation, virulence factor production, antibiotic resistance, and bioluminescence. By synchronizing gene expression, quorum sensing enables bacteria to behave as a unified group, enhancing their survival and adaptability in diverse environments [16,17]. Emerging evidence suggests that sRNAs play a key role in the regulation of quorum sensing [53,54]. This sRNA-mediated communication extends beyond intra-species interactions, as these regulatory molecules can be transferred between bacterial species and even across domains, impacting host–microbe interactions [55,56]. Understanding the mechanisms by which sRNAs mediate interspecies communication will provide deeper insights into microbial ecology and host-associated microbial dynamics, potentially informing strategies to manipulate microbial communities for therapeutic, agricultural, and industrial applications. This review will focus on the role of bacterial sRNAs in host–pathogen interaction by governing essential processes such as stress response, virulence regulation, antibiotic resistance, and interspecies communication, as outlined in the following sections.

## 2. sRNA as Orchestrators of Bacterial Pathogenesis

Pathogenic bacteria acquire foreign genes through horizontal gene transfer that can be integrated into the bacterial genome (pathogenicity islands) or exists as plasmids [57]. These horizontally acquired genetic elements harbor genes involved in bacterial pathogenicity, immune evasion, and persistence. These genetic elements are also a rich hub for small RNAs [58]. sRNAs play a crucial role in bacterial pathogenesis by precisely regulating the spatiotemporal expression of bacterial genes, ensuring adaptation to dynamic host environments, as detailed below.

Attachment to the host is the first step in bacterial colonization, allowing pathogens to anchor to host surfaces and establish an infection. Bacterial fimbriae, pili, and adhesins mediate these interactions, with sRNAs playing a crucial role in regulating their expression. For example, FimR2 sRNA controls the expression of fimbrial operons in *Escherichia coli* [59]. Similarly, small RNAs MicC and InvR regulate outer membrane protein OmpD in *Salmonella enterica* serovar Typhimurium and *S. entertidis*, respectively, that enables bacterial adherence to macrophages [60]. This regulation ensures optimal colonization while preventing premature immune recognition. Upon adherence, some bacteria deploy invasion mechanisms, often involving secretion systems that inject effector proteins into host cells. For example, *S. enterica* utilizes Type III Secretion Systems (T3SS) located at Salmonella pathogenicity island I (SP1), enabling epithelial cell invasion and intracellular survival [61]. sRNAs modulate the expression of these virulence genes located at SP1 in response to host-derived signals, ensuring their activation only when needed [62,63]. Following invasion, bacterial pathogens must evade immune responses and establish a replicative niche. For example, Mtb employs sRNAs to regulate dormancy and macrophage survival, facilitating long-term persistence [64]. Similarly, in *Salmonella*, sRNAs govern intracellular survival within macrophages by fine-tuning stress response pathways and modifying virulence gene expression [65]. Additionally, sRNAs contribute to iron uptake, quorum sensing, and biofilm formation, all of which are essential for bacterial persistence and dissemination, as described in detail in this section.

### 2.1. Adherence to the Host

Uropathogenic *Escherichia coli* (UPEC) encode at least nine types of fimbriae including type 1, P, S-fimbriae, and afimbrial adhesin I, which are encoded by the *fim*, *pap*, *sfa*, and *afa* operons, respectively [66,67] (Appendix A). Among these, type 1 fimbriae are the most prevalent, with more than 90% of both commensal and pathogenic *E. coli* strains harboring this adhesin [68]. In contrast, P-fimbriae, S-fimbriae, and afimbrial adhesin I are present in approximately 40–60%, 30–60%, and 0–12.5% of *E. coli* isolates, respectively [69]. Type 1 fimbriae play a crucial role in UPEC pathogenesis, as they facilitate adherence to and invasion of bladder epithelial cells, allowing intracellular replication and the formation of intracellular bacterial communities (IBCs) [12,13,14]. These fimbriae, encoded by the *fimAICDFGH* operon, play a crucial role in bacterial colonization, particularly in murine models of urinary tract infection (UTI). The expression of *fim* is regulated by multiple transcription factors, including Histone like Nucleoid Structuring protein, H-NS, Factor for inversion stimulation, Fis, and Leucine responsive regulatory protein, Lrp. Additionally, a 314-base pair inverted repeat element located upstream of *fimA* functions as a phase-variable switch (*fimS*), existing in either an ON or OFF state, controls the expression of the fimbrial operon. The inversion of the *fimS* switch is mediated by site-specific recombinases such as FimB and FimE, as well as IpuA, IpuB, and IpbA [70]. P-fimbriae enhance the virulence of UPEC by facilitating adhesion to uroepithelial cells in both in vitro and in vivo settings. Uropathogenic *Escherichia coli* (UPEC) is the primary causative agent of urinary tract infections (UTIs); enters the host through the urethra, ascends to the urinary bladder causing cystitis [71], and if left untreated, may progress to the kidneys, leading to pyelonephritis [72]. UPEC employs diverse virulence factors, including fimbrial adhesins, to colonize host tissues and evade immune defenses. In various Gram-negative bacteria, the chaperone–usher pathway (CUP) functions as a molecular machine that facilitates the assembly of fimbriae on the bacterial outer membrane; the adhesins located at the fimbrial tips play a crucial role in bacterial attachment to host epithelial receptors. UPEC strains encode multiple CUP operons. Several CUP-encoded fimbriae, including Auf, Dr, F1C, S, type 9, type 3, type 1, and P-fimbriae, have been identified in UPEC pathotypes. However, among those broadly conserved in UPEC isolates, only type 1 and P-fimbriae have been strongly associated with UTI pathogenesis [73]. Among these, P-fimbriae play a critical role in kidney colonization by binding to globoseries glycosphingolipids on uroepithelial cells [74]. Unlike type 1 fimbriae, which mediate mannose-sensitive adhesion, P-fimbriae facilitate mannose-resistant adherence, promoting persistent infection, resistance to urinary flow, and immune evasion [66]. A recent study has elucidated the role of small RNA PapR in the regulation of P-fimbriae [75], as discussed below.

#### 2.1.1. PapR

A large-scale phylogenomic analysis of 907 *E. coli* isolates, including 722 UPEC strains, identified the *papGII* locus, encoding P-fimbriae, as a key genetic determinant distinguishing invasive UPEC associated with severe UTI pathogenesis, such as pyelonephritis and urinary-source bacteremia, from non-invasive UPEC strains linked to asymptomatic bacteriuria or cystitis [76]. Notably, the number of *pap*-operon copies (*papGFEKJDCHABI*) varies among UPEC strains, with UTI89 harboring a single copy and CFT073 possessing two copies. Beyond their role in adhesion, P-fimbriae contribute to biofilm formation, enhancing bacterial survival by providing protection against host immune responses and antimicrobial agents [77]. The phase variation in P-fimbriae is tightly regulated through an antagonistic interaction between PapI and the small RNA PapR (Figure 1). PapR, a 180-nt long sRNA was 6-fold upregulated in the UPEC strain UTI89 during infection [75]. This small RNA is found in widely studied UPEC strains CFT073 and UTI89 and in some extraintestinal pathogenic *E. coli* (ExPEC) strains but is missing from *E. coli* K-12 strains. PapR expression was shown to be regulated by Lrp as deletion of *lrp* in UTI89 abrogated the expression of PapR [75].

Agglutination assays with yeast and human erythrocytes indicated that *papR* deletion increased mannose-resistant fimbrial expression, such as P-fimbriae, which was restored to wild-type level in the complemented strain [75]. Similarly, adhesion assays using bladder and kidney epithelial cells showed increased mannose-resistant adhesion in UTI89*ΔpapR*, reinforcing PapR’s role in regulating fimbrial expression [75]. Flow cytometry further confirmed that *papR* deletion doubled the fraction of P-fimbriated cells, while type-1 fimbriae levels remained unchanged [75]. These findings suggest that PapR negatively regulates P-fimbrial expression at both the population and single-cell levels. To achieve this, PapR turns the pap-operon to the OFF state by interacting with PapI mRNA, thereby preventing unnecessary fimbrial expression. While PapI promotes the ON state by interacting with the Lrp-DNA complex, PapR inhibits PapI by binding to a specific region (+74 to +96 nucleotides downstream of the PapI translational start site) (Figure 1a). This likely facilitates RNase recruitment or modulates ribosomal loading to repress PapI translation. Consequently, this inhibition disrupts the PapI-mediated translocation of the Lrp-DNA complex, thereby stabilizing the OFF phase [75].

In addition to PapR, the expression of P-fimbriae is subjected to epigenetic regulation by differential methylation of two GATC sites within the 400 bp Pap regulatory region. Methylation at the distal GATC site maintains repression, whereas methylation at the proximal site promotes the ON phase by enabling PapI function [75] (Figure 1a). Stress-responsive small RNA GcvB further modulates Lrp expression, while global regulators, including the cAMP receptor protein (CRP) and H-NS, impact P-fimbriae expression, with H-NS acting as a niche-specific repressor. The role of PapR extends beyond P-fimbriae silencing, as UPEC typically expresses only one fimbrial type at a time, leading to regulatory crosstalk [78]. The suppression of P-fimbriae by PapR may facilitate the compensatory upregulation of alternative adhesins, such as F1C fimbriae, ensuring adhesion plasticity. Together, PapR functions as a key regulatory checkpoint, restricting P-fimbriae expression to conditions where they are most beneficial for UPEC pathogenesis.

#### 2.1.2. RyfA

In addition to PapR, P-fimbriae are regulated by another sRNA called RyfA. It is a 305-nt sRNA implicated in oxidative and osmotic stress responses, biofilm formation, and swarming motility in *E. coli* and *S. dysenteriae* [79,80]. In UPEC strain CFT073, RyfA regulates genes involved in metabolism, respiration, and motility [79]. Interestingly, *Shigella dysenteriae* encodes two copies of RyfA namely RyfA1 and RyfA2, which are present in tandem in its genome and share about 95% in sequence identity [80]. RyfA1 but not RyfA2 was shown to be regulated by a second sRNA called RyfB1 and the overproduction of RyfA1 disrupts the virulence-associated process of cell-to-cell spread and reduces the expression of *ompC*, a key outer membrane protein essential for *Shigella* pathogenesis.

RyfA is crucial for *UPEC* pathogenicity by regulating fimbrial expression, modulating *fim* genes for type 1 fimbriae and *pap*-operons for P-fimbriae production. Additionally, *ryfA* mutants showed a 3.5-fold and 6.3-fold reduction in *papA* and *papA_2* expression and 2.02- and 3.16-fold reduction in *papC* and *papC2*, respectively, confirming its role in *pap*-operon regulation. Functional studies show that *ryfA* mutants exhibit significantly reduced fitness in a murine UTI model, with a 124-fold decrease in bladder colonization and a 13-fold reduction in kidney colonization compared to the wild-type strain [79]. These findings suggest RyfA’s role in adhesion and UPEC virulence.

#### 2.1.3. LhrC

LhrC is a multicopy small RNA in *Listeria monocytogenes* that is activated in response to cell envelope stress and infection-relevant conditions [81,82]. There are five variants of LhrC (LhrC1–5), each ranging from 111 to 114 nucleotides in length. While *lhrC1–4* is clustered together in the genome, *lhrC5* is located at a distant site. They are regulated by the stress-responsive two-component system (TCS) LisRK, linking their expression to environmental stressors such as low pH, ethanol exposure, and bile stress [81]. These sRNAs play a critical role in virulence regulation by targeting genes involved in bacterial adhesion and immune evasion. Bioinformatics and *β*-galactosidase assays identified *lapB* mRNA, encoding virulence-associated adhesin, as a direct target of LhrC1–5. Akin to PapR, which negatively regulates the expression of P-fimbriae discussed above, LhrC negatively regulates *lapB* expression under bile and antibiotic stress or in blood, facilitating immune evasion by downregulating adhesin expression [81]. Unlike redundant regulatory mechanisms often seen in multicopy sRNAs, LhrC1–5 exhibit additive activity, with their CU-rich motifs binding to the AG sequence within the ribosomal binding site of target mRNAs, allowing precise posttranscriptional regulation. Transcriptomic and proteomic analyses further revealed that LhrC regulates additional virulence-associated genes, including the oligopeptide-binding protein *oppA*, the amino acid ABC transporter *lmo2349*, and the CD4⁺ T-cell-stimulating antigen TcsA. Interestingly, LhrC variants demonstrated the ability to bind two OppA target mRNAs via two of their three CU-binding sites, suggesting a sophisticated regulatory mechanism [82]. These findings provide novel insights into LhrC-mediated gene regulation and highlight the potential of targeting LhrC for antimicrobial strategies aimed at disrupting *Listeria* virulence.

#### 2.1.4. STnc640

STnc640 is an sRNA that plays a key role in adhesion and virulence regulation in *Salmonella* Enteritidis strain 50336. Sequence analysis revealed that it shares 97% homology to its counterpart in *Salmonella* Typhimurium [83]. Using bioinformatics and RT-qPCR, Meng et al. demonstrated that STnc640 represses *fimA*, the major structural gene encoding type 1 fimbriae, via an Hfq-mediated posttranscriptional regulatory mechanism [60]. The deletion of *STnc640* enhanced bacterial adhesion to Caco-2 cells and increased virulence in a chicken infection model, suggesting that STnc640 acts as a negative regulator of adhesion-related genes. Future studies, including overexpression experiments, are necessary to confirm its precise role in regulating fimbrial expression and virulence attenuation in *Salmonella* Enteritidis.

### 2.2. Invasion of the Host

A key attribute of successful pathogens is their ability to invade the host, facilitated by various virulence factors. For example, UPEC produces α-hemolysin, encoded by the *hlyCABD* operon, which disrupts host cell membranes, causing lysis and tissue damage, thereby enhancing bacterial colonization of the urinary tract [84,85]. Additionally, α-hemolysin modulates host immune responses by suppressing pro-inflammatory cytokine expression, allowing UPEC to evade immune detection. Similarly, *Staphylococcus aureus* secretes Protein A, which binds to the Fc region of antibodies, preventing opsonization and phagocytosis [86]. *Bordetella pertussis* produces adenylate cyclase toxin, increasing cyclic AMP levels in host cells and impairing immune function [87], while *P. aeruginosa* releases Exotoxin A, which inhibits protein synthesis and induces immune cell death [88] (Appendix A). Furthermore, *Clostridiodes difficile* produces Toxins A and B, which disrupt the cytoskeleton of intestinal cells, leading to inflammation and immune evasion [89]. These bacterial toxins employ diverse strategies to subvert host defenses, facilitating survival and pathogenesis. The following section highlights the role of bacterial small RNAs in regulating virulence and host invasion, adding another layer of complexity to bacterial pathogenicity. Shiga toxin (Stx) is a major virulence factor of *S. dysenteriae* and enterohemorrhagic *Escherichia coli* (EHEC) [90]. It is an AB5 toxin, comprising an enzymatically active A subunit and a pentameric B subunit that binds globotriaosylceramide (Gb3) receptors on host cells. Upon internalization, the A subunit inhibits protein synthesis by cleaving an adenine residue from 28S rRNA, leading to cell death, epithelial damage, inflammation, and hemorrhagic colitis. In severe cases, Stx enters the bloodstream, causing endothelial damage in the kidneys and triggering hemolytic uremic syndrome (HUS), characterized by acute kidney injury, thrombocytopenia, and hemolytic anemia [91]. The toxin’s genes, carried by lambdoid bacteriophages, enable horizontal transfer of virulence factors, driving bacterial evolution [92]. Stx is categorized into two primary groups, Stx1 and Stx2, with subtypes such as Stx1a, Stx2a, and Stx2d associated with severe disease.

#### 2.2.1. StxS

Shiga toxin 1 (Stx1) expression is regulated through three pathways: (1) the phage late promoter (PR’), activated by phage induction and the SOS response, integrates signals from quorum sensing, antibiotics, and inhibitors [93]; (2) a Fur-responsive promoter specific to Stx1, modulating expression based on iron and nitric oxide levels [94]; and (3) posttranscriptional regulation by sRNAs [43]. During the lytic cycle, toxin genes are embedded within the late transcript of the phage and are regulated via antitermination of the PR′ late promoter. In the lysogenic state, transcription from PR′ is prematurely terminated at tR′, resulting in the production of a truncated 74-nt RNA, StxS as a byproduct of antitermination control. StxS represses Stx1 toxin production by binding to the ribosome binding site of *stx1B*, silencing its translation [43] (Figure 1c). Interestingly, StxS does not regulate *stx2AB*, which is exclusively induced during the lytic cycle, suggesting that other Hfq-dependent sRNAs may modulate Stx2a expression. This posttranscriptional regulation ensures minimal Stx1 production during lysogeny, acting as a safeguard. In addition to repressing Stx1, StxS activates the general stress response regulator RpoS by binding to its 5′ untranslated region (UTR), relieving repressive secondary structures and preventing Rho-dependent termination (Figure 1c). This activation enhances bacterial stationary phase survival, increasing cell density under nutrient-limited conditions by approximately 20%. Another sRNA, DicF, was shown to regulate Shiga toxin. As DicF responds to changes in oxygen level (the role of DicF is discussed in detail in the bacterial stress response section), the pathogen can modulate the expression of the toxin by sensing the oxygen tension across different regions of the gut [95].

#### 2.2.2. RNA III

*Staphylococcus aureus* α-hemolysin (Hla), a β-barrel pore-forming toxin, is essential to the pathogen’s virulence [96]. Secreted as a monomer, Hla oligomerizes into a heptameric pore on host cell membranes, forming a 1–3 nm β-barrel pore that disrupts cellular ionic homeostasis and facilitates the passage of small molecules. At lytic concentrations, Hla causes cell lysis, while sublytic levels alter host signaling, affecting processes such as proliferation, inflammation, cytokine secretion, and cell–cell interactions [97]. Hla targets diverse human cell types, including epithelial, endothelial, and immune cells. Its activity is regulated by the Agr quorum sensing system and involves interactions with host lipids [98], contributing to species- and cell type-specific effects [96]. Structural and mechanistic insights have informed therapeutic strategies aimed at mitigating Hla-mediated diseases.

The quorum sensing system in *S. aureus* encodes two divergent RNAs: RNAIII, a central effector, and RNAII, encoding the *agr*-operon [98]. RNAIII, 514-nt sRNA upregulates Hla expression through two distinct mechanisms. First, it binds to the 5′ UTR of Repressor of toxin (Rot) mRNA, a SarA ortholog that represses *hla* and other virulence factors, such as *hlgB* and *hlgC* (encoding gamma-hemolysin components) and *sspB* and *sspC* (encoding cysteine protease precursors) [99]. By silencing Rot, RNAIII promotes toxin and virulence factor expression. Second, RNAIII directly binds *hla* mRNA, exposing the Shine–Dalgarno (SD) sequence and enhancing translation [100].

Recent studies identified SprY, an sRNA encoded by the prophage φ12, as a novel virulence regulator through its interaction with RNAIII. It acts as an RNA sponge, binding the 13th stem-loop of RNAIII—the same site targeted by multiple RNAIII-regulated mRNAs, such as *rot*, *spa*, and *eca* [101]. This interaction prevents RNAIII from modulating its targets, enhancing Rot translation and reducing hemolytic activity. SprY is most active during early growth phases, facilitating colonization by suppressing toxin production. Conversely, SprX, another RNAIII-binding sRNA, exerts opposing effects, illustrating a complex network of sRNA interactions in fine-tuning virulence factor expression.

Additionally, Morales-Filloy et al. identified 5′-NAD-capped RNAs in *S. aureus*, including RNAIII, using NAD capture sequencing. RNAIII exhibits variable NAD modification levels [102]. NAD capping reduces α- and δ-hemolysin production and alters the expression of the *splABCD* operon, which is involved in tissue dissemination. Although in vitro studies suggest that NAD capping does not significantly affect RNAIII’s secondary structure or translation, it may modulate regulatory functions in vivo by interacting with other factors. NAD capping is thought to fine-tune bacterial responses to redox state and quorum sensing signals. NAD^+^/NADH 5′ modification of RNA occurs across bacteria, eukaryotes, and viruses, with RNA polymerases incorporating NAD during synthesis initiation. For example, in *E. coli*, RNA NADylation is enhanced under low ATP conditions, weak ATP-RNA polymerase interactions, or DNA supercoiling [103]. Although NADylated RNA remains susceptible to RNase E processing and translation, it undergoes rapid degradation by NudC decapping, a process inhibited by the stringent response molecule ppGpp. This inhibition stabilizes NADylated RNAs during nutrient limitation, extending RNA half-life and facilitating efficient protein expression under stress conditions. In *S. aureus*, this mechanism may provide an energy-efficient strategy for ensuring stable toxin production and stress response under host-induced nutrient limitations.

#### 2.2.3. BtsR1

Bacteria sRNA can also be managers of resource optimization ensuring the timely expression of their target only in the presence of the stimuli as exemplified by the *Bacillus thuringiensis* strain YBT-1518. This strain harbors three nematicidal *cry* genes (*cry55Aa*, *cry6Aa*, and *cry5Ba*), with high toxicity against *Caenorhabditis elegans* [104]. The 51-nt sRNA BtsR1 directly regulates *cry5Ba* expression by binding to the ribosome binding site (RBS) of Cry5Ba mRNA, preventing translation. During free-living conditions, the suppression of Cry5Ba by BtsR1 reduces toxin production, minimizing energy expenditure and avoiding host detection. Interestingly, once the bacterium is ingested by *C. elegans*, BtsR1-mediated silencing is alleviated, leading to the activation of Cry5Ba toxin expression within the host. The toxin then subverts the nematode’s defenses and ultimately causes its death. This dynamic regulatory mechanism enables the bacterial pathogen to optimize toxin production in response to environmental conditions. Cry proteins are energetically costly, constituting over 25% of the bacterial dry mass, and their production during free-living conditions would waste resources. By silencing toxin synthesis via BtsR1, YBT-1518 avoids triggering host avoidance behaviors and ensures efficient ingestion by nematode hosts. Once inside the host, the reduced transcription of BtsR1 allows the bacterium to activate Cry5Ba production, securing nutrients critical for its survival. This mechanism of toxin silencing highlights the ecological significance of sRNA-mediated regulation in balancing energy conservation with survival strategies. BtsR1 exemplifies how regulatory sRNAs enable pathogens to fine-tune gene expression in response to environmental and host-derived signals. This sRNA-mediated strategy reflects the co-evolutionary arms race between *B. thuringiensis* and nematode hosts [104].

### 2.3. Evading Immune Response

The sRNAs discussed in the previous sections also contribute to the growth of bacterial pathogens inside the host and at the same time equip the bacterial pathogen with defense to evade the host immune response. For example, RyhB and OxyS are known to regulate iron homeostasis and oxidative stress response to mitigate host immune defense and thus have emerged as major players of virulence in pathogenic strains of *E. coli* [105,106,107], *Vibrio* [108], *Salmonella* [65], and other Gram-negative bacteria. Similarly, bacteria have evolved diverse strategies to evade the host immune response, enabling their survival and persistence within the host [12,14]. One common approach is the modulation of surface structures to avoid recognition by the host’s immune system. For instance, altering or masking pathogen-associated molecular patterns (PAMPs), such as lipopolysaccharides or flagellin, prevents the activation of Toll-like receptors (TLRs) on immune cells [109,110]. Bacteria also secrete effector proteins through specialized secretion systems (e.g., Type III and Type IV secretion systems) to manipulate host cell signaling pathways, suppress pro-inflammatory responses, and inhibit apoptosis [61,111] (Appendix A). Additionally, some pathogens form biofilms, a protective extracellular matrix that shields them from phagocytosis and antimicrobial agents [112]. Intracellular pathogens like *Salmonella* and *Listeria* exploit host cells by residing in modified vacuoles or escaping into the cytosol, thus avoiding lysosomal degradation [113,114]. There is growing evidence about the role of sRNAs in immune evasion. For example, sRNAs modulate the expression of virulence factors that interfere with cytokine production, or inhibit oxidative stress pathways to enhance bacterial survival [115]. These multifaceted evasion tactics underscore the sophisticated interplay between bacteria and host immune defenses, as discussed in the following examples.

#### 2.3.1. FasX

Group A *Streptococcus* (GAS) such as *Streptococcus pyogenes* is a human pathogen responsible for a broad spectrum of diseases including streptococcal pharyngitis (strep throat) and the highly invasive and often deadly necrotizing fasciitis (flesh-eating syndrome) [116,117]. It produces a 205-nt sRNA, FasX, that plays a major role in immune evasion of the pathogen. It is regulated by the FasBCA system, with serotype-specific differences in basal expression potentially linked to single nucleotide polymorphisms (SNPs) in the *fasX* promoter. FasX reduces pilus expression by binding to the pili mRNAs inhibiting their translation [118] (Figure 1d). GAS pilus expression is tightly regulated, as it can enhance adhesion and biofilm formation but also increase bacterial susceptibility to neutrophil extracellular traps [118]. The interaction between FasX and its target pili gene is serotype-specific. For example, in serotypes M1 and M2, FasX targets genes encoding the minor pilus protein (M1.cpa and M2.113), while in serotype M6 and M28, it regulates genes encoding the major pilus protein (M6.tee6 and M28.fctA). FasX also negatively regulates the expression of the fibronectin-binding proteins PrtF1 and PrtF2, which promote adhesion and internalization. FasX represses PrtF1/2 expression through posttranscriptional regulation by base pairing with the PrtF1 and PrtF2 mRNAs within their 5′UTR, specifically overlapping the ribosome binding sites. This duplex formation impedes ribosome access, thereby inhibiting mRNA translation [45].

Conversely, FasX increases streptokinase (a plasminogen activator that produces serine protease plasmin, which promotes bacterial metastasis) expression by stabilizing its mRNA [119]. These regulatory actions of FasX contribute to the pathogen’s ability to cause disease by contributing to the transition from colonization to dissemination across multiple GAS serotypes, as demonstrated in a bacteremia model using human plasminogen-expressing mice [119]. Reduced pilus expression decreases GAS adherence to host tissues, while increased streptokinase expression promotes tissue barrier breakdown and immune evasion (Figure 1d).

#### 2.3.2. MTS1338

In Mtb, sRNA MTS1338 plays a crucial role in virulence and intracellular survival [39,120]. MTS1338, highly conserved across pathogenic mycobacteria, is upregulated in dormant Mtb and macrophage-phagocytosed cells, suggesting a role in host–pathogen interactions. Elevated MTS1338 expression during infection is linked to transcriptional changes indicative of reduced metabolism and cell division, strategies associated with stress survival. Transcriptomic analyses revealed a characteristic response to MTS1338 overexpression during logarithmic and stationary phases, with increased resistance to low pH, reactive oxygen species (ROS), and reactive nitrogen species (RNS) in logarithmic-phase [121]. Survival analysis demonstrated enhanced stress resistance, highlighting MTS1338’s role in adapting to the macrophage environment. Interestingly, approximately 50% of differentially expressed genes (DEGs) in MTS1338-overexpressing strains overlapped with those activated in Mtb residing within macrophages, underscoring its importance in intracellular adaptation. These include MPT70, encoding an antigen involved in T-cell differentiation and expressed in later infection stages, Rv2034 and Rv2035, components of a toxin–antitoxin system and regulators of virulence and stress responses, and CmtR (Rv1994c) and Rv2642, ArsR-family regulators linked to ROS detoxification and metal stress (Figure 1e).

MTS1338 is regulated by DosR, a transcriptional activator under hypoxic conditions, although it is not fully dependent on DosR [122,123]. The DevR/DosR response regulator plays a crucial role in the virulence, dormancy adaptation, and antibiotic tolerance of Mtb by regulating the dormancy regulon. Previously, we developed recombinant tools to study gene regulation in Mtb [124,125]. Using this in vivo reporter assay, we characterized key residues involved in DevR–RNA polymerase interactions. In short, this system utilizes *E. coli* co-transformed with plasmids expressing Mtb RNA polymerase subunits (α, β, β′, and σ^A^), a DevR expression cassette, and a GFP reporter under the DevR-regulated *fdxA* promoter. We found that the amino acid E280 in the α-subunit of RNAP and K513 and R515 in sigma factor A are involved in DevR-mediated transcription. In silico modeling suggests that interactions between key residues of DevR (D172, E178) and sigma factor A (Q505, R515, K513) are required for DevR-mediated transcriptional regulation [126]. MTS1338 upregulation has been observed in hypoxic, acidic, and oxidative stress conditions, suggesting a role in dormancy [120]. Biochemical studies indicate that DosR binds with high affinity to MTS1338′s promoter region, enhancing its expression [127]. Although PhoP does not directly bind to the MTS1338 promoter, it interacts with the DosR-MTS1338 complex, indicating a coordinated regulation mechanism. Taken together, these findings underscore MTS1338’s importance in Mtb pathogenicity, facilitating adaptation to macrophage-induced stress and promoting bacterial survival during chronic infection.

It is thus clear that bacteria have evolved a diverse repertoire of sRNAs to regulate virulence factor expression, facilitating host infection and immune evasion. Beyond virulence regulation, sRNAs also contribute to complex stress response pathways, enabling bacterial adaptation to various host-induced stresses. The following section explores the role of sRNAs in mediating stress responses to counteract host defense mechanisms.

## 3. Role of Small RNA in Stress Response

A bacterial pathogen faces a multitude of hostile conditions that constitute significant stress within a host [32]. Immediately upon entry, the pathogen encounters the host’s innate immune defenses, including antimicrobial peptides and reactive oxygen and nitrogen species produced by phagocytes. In addition, it is faced with host-mediated nutrient limitation (nutritional immunity). Furthermore, bacterial pathogens are subjected to further stresses like acidic pH within phagosomes, lysosomal enzymes, and continued oxidative and nitrosative attack [128]. Beyond direct immune assault, pathogens must also adapt to the limited availability of essential nutrients like iron and specific carbon sources, competition with the host microbiota, along with fluctuations in temperature and osmolarity [129,130]. These diverse and dynamic stresses necessitate sophisticated bacterial adaptation mechanisms to ensure survival and successful infection.

Bacteria have evolved a diverse arsenal of defense mechanisms to counteract the stresses imposed by the host [32]. To combat ROS and RNS, bacteria employ antioxidant enzymes like superoxide dismutase [131,132], catalase [133], and alkyl hydroperoxide reductase [134]. To survive within acidic phagosomes, bacteria utilize acid tolerance systems, including proton pumps to maintain intracellular pH [135] and urease to generate ammonia [136]. Additionally, certain bacteria form biofilms, providing a protective matrix against immune cells and antimicrobial agents [112]. Furthermore, certain pathogens can modulate host immune responses directly, inhibiting phagosome maturation, escaping into the cytoplasm, or suppressing cytokine production, effectively manipulating the host’s defenses to their advantage. Finally, nutrient limitation is addressed through metabolic flexibility, allowing bacteria to utilize alternative carbon sources and specialized scavenging and transport systems for essential nutrients like iron [137]. Iron is a double-edged sword for living organisms. While indispensable for essential processes like respiration, DNA synthesis, and enzymatic catalysis, excess iron can be toxic due to the formation of ROS through the Fenton reaction. Therefore, maintaining iron homeostasis is critical for bacterial survival, especially within host environments where iron is often scarce due to host nutritional immunity. In *E. coli*, the transcription factor Ferric Uptake Regulator, Fur, acts as a master regulator of iron uptake. Fur regulates a small RNA, RyhB, that equips the pathogen with a stress response, as outlined below.

### 3.1. RyhB

RyhB is a 95 nt sRNA that plays a key role in iron homeostasis and virulence in several pathogenic and non-pathogenic strains of bacteria including *E. coli* [138], *Salmonella* [139], and *Vibrio* [108] (Figure 2). It regulates targets through translational repression and mRNA degradation, often mediated by Hfq and RNase E. These include genes involved in the TCA cycle, respiration, and siderophore synthesis. RyhB is directly repressed by Fur [140]. Under iron-limiting conditions, RyhB was shown to play an iron-sparing response by inhibiting the translation of non-essential protein that uses Fe or Fe-S as a co-factor, thereby ensuring that the essential proteins can ligate to Fe-containing co-factors to drive key metabolic processes needed for bacterial survival [140] (Figure 2a,b).

Previous works have suggested that the deletion of *fur* did not impact virulence, but that the loss of *ryhB* or both *fur* and *ryhB* impaired bladder colonization in the murine model of infection [105]. Although the role of Fur and RyhB have been well characterized in the non-pathogenic strains of *E. coli*, their contribution to the pathogenic strains remained elusive. To address this, using genomic and molecular biology tools like ChIP-seq, RNA-seq, and Proteomics, we have characterized the Fur and RyhB regulon in UPEC strain CFT073, revealing a complex interplay among iron acquisition, stress responses, and metabolic adaptation [140]. Fur directly and indirectly regulates a wide array of genes, including those involved in siderophore biosynthesis, iron uptake, and stress mitigation pathways, with RyhB modulating some components posttranscriptionally. Unique to CFT073, Fur-regulated pathways such as the siderophore aerobactin biosynthesis impose a metabolic burden, leading to increasing amino acid demand, which may be mitigated by external amino acid uptake. We found that the absence of Fur, a condition that mimics an Fe-limiting environment, induced the RpoS-mediated stringent stress response pathways in CFT073. This activation of the stringent stress pathways may provide the pathogen with an anticipatory response, where iron limitation could also be an indicative of other nutrient limitation, as the pathogen is often encountered in the nutrient-limited urinary tract (Figure 2b). This regulatory network, distinct from that in commensal strain MG1655, underscores the role of Fur and RyhB in optimizing iron and other nutrient acquisition and virulence in nutrient-limited host niches, offering insights into UPEC pathogenesis and potential targets for controlling infections [140]. Additionally, RyhB was also shown to play a key role in virulence of an avian pathogenic *E. coli* strain by regulating the expression of genes involved in biofilm formation, adhering to brain microvascular endothelial cell line, and contributing to meningitis development in the murine model of infection [106].

Recent work has identified the role of RyhB in persistor formation [141]. Persistor formation is one of the key strategies employed by bacteria to convert itself into a metabolically inactive state to evade the host immune response (discussed in detail in the antibiotic section). Interestingly, *Salmonella* contains two copies of *ryhB* in the genome *ryhB-1* and *ryhB-2*, which are encoded by separate genes found across all sequenced *Salmonella* species [139]. These sRNAs share a conserved 33-base region, while the remainder of their sequences exhibit lower conservation. RyhB-1 and RyhB-2 have both overlapping and unique regulatory roles. As in *E. coli*, the expression of both *ryhB-1* and *ryhB-2* is repressed by Fur; however, their activation is context-dependent. RyhB-1 is induced by oxidative stress caused by H₂O₂, whereas RyhB-2 is activated during the stationary growth phase. As in *Salmonella*, sp., Acuña et al. characterized two RyhB homologs, RyhB-1 and RyhB-2, in the fish pathogen *Yersinia ruckeri* [142]. Both homologs are induced under iron starvation, repressed by Fur, and stabilized by Hfq, similar to other Enterobacteriaceae. However, their functional roles differ. The *ΔryhB-1* mutant exhibited increased motility, reduced biofilm formation, faster growth, and higher ATP levels, along with enhanced survival in salmon cell cultures. In contrast, the *ΔryhB-2* mutant was non-motile and formed more biofilm. Gene expression analysis revealed that RyhB-1 and RyhB-2 regulate multiple genes including transcriptional regulators HilA, HilC, HilD, InvF, and RtsA, as well as the regulatory protein RtsB. Additionally, it influences the invasion chaperone SicA, the needle proteins PrgI and InvI, and the effector proteins SipA, SipB, and SipC. Furthermore, RyhB affects the expression of SPI-1-associated genes encoding the effector proteins SopA, SopB, and SopD and tricarboxylic acid (TCA) cycle genes *fumA* and *sdhD*.

Interestingly, Coronel-Tellez et al. identified the small RNA IsrR that provides an iron-sparing response in *Staphylococcus* sp. akin to RyhB [143]. IsrR is repressed by iron, and it downregulates the expression of mRNAs encoding iron-containing enzymes, particularly those involved in anaerobic nitrate respiration. Structural analysis revealed three C-rich regions (CRRs) in IsrR, each contributing differently to target regulation. IsrR is essential for full virulence in a mouse septicemia model, highlighting its role in bacterial survival during infection. Although not orthologous to RyhB or other known sRNAs, IsrR regulates common targets, demonstrating convergent evolution in bacterial iron-sparing responses.

### 3.2. MsrI and Mcr11

MrsI (ncRv11846) is a 100-nt sRNA identified in *Mycobacterium* sp. [144]. MrsI is regulated by the iron-dependent transcription factor IdeR, in both *M. tuberculosis* and *M. smegmatis*, driving its expression under iron-limiting conditions. High-throughput sequencing and differential expression studies identified MrsI as significantly upregulated during iron deprivation and membrane stress. Additionally, the CRISPR-mediated knockdown of MrsI further confirmed its regulatory role in iron homeostasis, with 118 genes upregulated including bacterioferritin, hydrogenase maturation factor (HypF), and ferredoxin reductase (FprA). Functionally, MrsI represses *bfrA* mRNA, encoding bacterioferritin, an essential iron storage protein [144]. This repression is mediated by a conserved six-nucleotide seed sequence and a seven-nucleotide apical loop. Collectively, these findings highlight MrsI as a critical regulator of iron metabolism, essential for Mtb adaptation to iron-limiting environments, akin to RyhB in *Escherichia coli*.

Another sRNA, Mcr11, in Mtb, plays a crucial role in stress responses during slow growth and chronic infection. It is regulated by the transcription factor AbmR [145] by a mechanism that involves slowing down the transcribing RNA polymerase by binding to both RNAP and the DNA, similar to that observed for Rv1222 [146]. Bioinformatic predictions and qRT-PCR analyses identified *Rv3282*, *fadA3*, and *lipB* as direct targets of Mcr11, with base pairing sites located immediately upstream of these genes. Notably, *lipB* expression was significantly de-repressed in Mcr11-deficient Mtb, confirming its regulatory role. Furthermore, Mcr11-mediated gene regulation was influenced by the presence of fatty acids, linking it to central metabolism. These findings highlight the importance of Mcr11 in Mtb’s adaptation and underscore the need to further explore sRNA stability and function in tuberculosis pathogenesis.

### 3.3. OxyS

OxyS is a 117-nt sRNA in *Escherichia coli* that plays a central role in the oxidative stress response [147]. Its expression is induced by the transcription factor OxyR under oxidative stress conditions. OxyS regulates the expression of approximately 40 genes [148] including genes involved in stress response, motility, DNA repair, and antibiotic resistance. One of the key functions of OxyS is to repress the transcriptional activator *fhlA*. Using Nuclear Magnetic Resonance (NMR), Small Angle X-ray Scattering, SAXS, and molecular modeling, Štih et al. revealed that OxyS adopts a boomerang-like conformation consisting of four stem-loops (SL1-SL4). The stem-loops SL1 and SL3 are positioned at opposite ends whereas SL4 shortens the unstructured linker to adopt a confirmation that allows the repression of *fhlA* [149]. The chaperone protein Hfq enhances *oxyS*-*fhlA* interactions by destabilizing SL2 and partially unfolding SL4, promoting more efficient target binding, although OxyS retains regulatory function in the absence of Hfq, possibly forming stable Watson–Crick base pairs with *fhlA*, with magnesium ions facilitating partial unfolding for enhanced base pairing. This structural plasticity likely underlies OxyS’s ability to regulate multiple targets. OxyS also negatively regulates the expression of the alternative sigma factor RpoS, which is essential for general stress resistance in *E. coli*. While initial studies proposed direct base pairing between OxyS mRNA and the RpoS leader sequence, later research demonstrated that OxyS has low affinity for the *rpoS* mRNA leader region despite its strong binding to Hfq [150]. This suggests an indirect mechanism, where OxyS sequesters Hfq, reducing its availability to other sRNAs, including DsrA, a known activator of *rpoS*.

The overexpression of OxyS inhibits swimming and swarming motility and reduces curli expression, linking it to biofilm formation [42]. Additionally, OxyS contributes to cellular stress adaptation by inducing temporary cell cycle arrest, allowing for DNA repair [151]. To achieve this, it represses the expression of transcription termination factor NusG, leading to the activation of the cryptic prophage gene *kilR*. KilR inhibits FtsZ, a key protein in bacterial cell division, slowing proliferation and allowing DNA repair to proceed effectively (Figure 2c). OxyS influences antibiotic susceptibility, particularly cephalothin resistance, by modulating target gene expression [152]. Transcriptomic and phenotypic analyses by Cho et al. identified 27 differentially expressed OxyS-regulated genes, including *cycA* (D-serine/alanine/glycine:H^+^ symporter) and *cysH* (phosphoadenosine phosphosulfate reductase). Knocking out 17 genes out of the 27 OxyS targets exhibited increased resistance. Nine were linked to the cAMP receptor protein (CRP), a global transcriptional regulator in *E. coli*. These findings suggest that OxyS modulates multiple pathways to confer stress response and may serve as a molecular marker or target for identifying and combating antibiotic resistant strains.

### 3.4. DsrA

There are two global regulators that play a key role in the stress response in *E. coli*. The first is the stress-induced sigma factor RpoS that equips the pathogen with necessary adaptations to cope with the stress. The second is H-NS, a nucleoid-associated protein in bacteria akin to histones in eukaryotic systems that binds across the genome and acts as a global transcriptional repressor, silencing genes involved in stress responses and virulence. Although several transcription factors regulate the expression of RpoS and H-NS, an 87-nt sRNA called DsrA regulates the expression of both RpoS and H-NS antagonistically. DsrA activates the expression of *rpoS* posttranscriptionally by binding to 5′ UTR and disrupting inhibitory secondary structures of RpoS transcript [153]. Alternatively, it represses the expression of *hns*, causing de-repression of H-NS silenced genes that are involved in stress response.

DsrA promotes acid resistance by activating genes involved in the acid stress response including the *hdeAB*, *gadAX*, and *gadBC* operons in *E. coli* K-12 strains, as well as those in the pathogenic *E. coli* strain O157:H7, contributing to bacterial endurance in acidic environments such as the stomach [154]. In *Salmonella enterica* serovar Typhimurium, DsrA influences central carbon metabolism (CCM) and NAD(H) homeostasis by altering the expression of genes involved in glycolysis, pyruvate metabolism, the TCA cycle, and NADH-dependent respiration. A GFP-fluorescence-based reporter assay identified *pflB*, encoding pyruvate-formate lyase, as a direct target of DsrA repression. DsrA base pairs with *pflB* mRNA, recruiting RNase E to degrade it, thereby reducing *pflB* stability. This repression lowers the NAD^+^/NADH ratio, helping maintain redox balance [155]. Additionally, DsrA plays a key role in prophage λ induction by relieving the repression of *rcsA* through antagonizing H-NS. This leads to increased capsular polysaccharide (*cps*) expression and RecA-independent prophage induction. Unlike RcsA, which requires RcsB for induction, DsrA functions independently of RcsB in stationary phase, suggesting an alternative pathway for prophage activation [156,157]. Recent findings suggest that DsrA also plays a role in replication initiation, requiring Hfq, Dps, and OxyR [158].

### 3.5. RybB

The outer membrane (OM) is crucial for the survival of Gram-negative bacteria. In *E. coli*, the sigma factor E, SigE, plays a crucial role in envelope stress responses by detecting signals that are indicative of OM dysfunction and thereby initiates an adaptive response by upregulating genes involved in the biogenesis, transport, and assembly of key OM components, including lipopolysaccharides (LPSs), phospholipids, and outer membrane proteins (OMPs) [159]. Additionally, SigE regulates the expression of proteases and chaperones that contribute to OM maintenance and repair, ensuring membrane integrity under envelope stress conditions. One key player in this stress response is a 79-nt sRNA called RybB. The expression of RybB is regulated by SigE and the stress-associated alarmone ppGpp in *E. coli* and *Salmonella* sp. [160]. Like other trans-acting sRNAs, RybB’s function depends on RNA chaperones, including Hfq and ProQ. Hfq stabilizes RybB and facilitates its interactions with target mRNAs, while ProQ protects it from degradation. Its key targets include mRNAs encoding outer membrane proteins such as *ompC*, *ompW*, *ompA*, and *fadL* [161] (Figure 2d). By targeting these mRNAs, RybB reduces the production of porins and fatty acid transporters, facilitating rapid adjustments to envelope composition. In *Salmonella*, RybB plays a greater role to cope with envelope stress by regulating additional targets like *ompD*, *ompF*, *ompN*, *ompS*, and *tsx*. This ability to modulate OMP expression is crucial for bacterial adaptation to stressors, including changes in osmolarity, pH, and exposure to antimicrobial agents. Interestingly Adams et al. identified FtsO, an ORF-internal sRNA derived from the *ftsI* mRNA, which plays a crucial role in bacterial cell division, acting as an RNA sponge to negatively regulate RybB. An RNA sponge is a regulatory RNA that sequesters and inhibits sRNAs by base pairing, preventing them from interacting with their target mRNAs. Since *ftsI* encodes a penicillin-binding protein essential for cell division, the regulation of RybB by FtsO suggests a mechanism linking membrane stress response with the division cycle [162]. RybB also acts as a negative regulator of biofilm formation by negatively regulating the expression of Curlin subunit gene D, CsgD. Beyond biofilm regulation, RybB promotes stationary phase cell lysis, a process critical for population turnover, eliminating damaged cells, and preventing mutation accumulation. Disrupting RybB or the stress-responsive sRNA, MicA impairs lysis, reducing survival and increasing mutation rates, ultimately causing population collapse.

In summary, bacterial small RNAs play a pivotal role in enabling pathogens to withstand various host-induced stresses, thereby contributing significantly to bacterial pathogenesis and host–pathogen interactions. Moreover, bacteria frequently encounter antibiotics, which are administered to eliminate pathogenic infections. The following section explores the role of small RNAs in mitigating the adverse effects of antibiotic exposure and enhancing bacterial survival.

## 4. Role of sRNA in Antibiotic Resistance

As discussed in the previous sections, bacterial pathogens pose a significant global threat due to their diverse virulence factors and stress-responsive mechanisms, which enable them to evade host immunity and establish infections. To combat these pathogens, antibiotics were developed as a crucial means of treatment [163]. However, in recent years, the efficacy of the antibiotics has gone down due to the rise in multi-drug-resistant strains [19]. Antibiotics enter bacterial cells through various mechanisms, depending on their class and the bacterial cell membrane/wall structure and target essential processes, thereby killing (or inhibiting) the pathogen. For instance, Beta-lactams (e.g., penicillins, cephalosporins) diffuse through the peptidoglycan layer in Gram-positive bacteria, inhibiting cell wall synthesis [164], whereas aminoglycosides (e.g., gentamicin, streptomycin) pass through porin channels in Gram-negative bacteria and bind ribosomal RNA, disrupting protein synthesis [165]. Similarly, Tetracyclines (e.g., doxycycline) utilize passive diffusion and active transport to inhibit ribosomal function [166]. Macrolides (e.g., erythromycin, azithromycin) diffuse through membranes to bind the 50S ribosomal subunit, halting protein synthesis [167]. Fluoroquinolones (e.g., ciprofloxacin) enter via porins and inhibit DNA replication by targeting DNA gyrase and topoisomerase IV [168]. Sulfonamides (e.g., sulfamethoxazole) mimic para-aminobenzoic acid (PABA) to block folic acid synthesis [169]. Glycopeptides (e.g., vancomycin) target cell wall precursors, preventing synthesis [170]. Taken together, each class of antibiotics has a unique mechanism of entry and action, which is crucial for their effectiveness and the development of resistance mechanisms.

To curb the effects of antibiotics, bacteria have utilized sRNAs to develop elaborate strategies that will be discussed in the following section. For example, treatment with last-line antibiotics, including linezolid, ceftobiprole, tigecycline, and vancomycin of methicillin-resistant *Staphylococcus aureus* (MRSA) exhibited a coordinated sRNA response [171]. Similar sRNA-dependent responses to sub-MIC levels [antibiotic concentrations below the minimum inhibitory concentration (MIC), meaning they are insufficient to prevent bacterial growth] of eight clinically relevant antibiotics and tigecycline was also observed in DOT-T1E isolate of *Pseudomonas putida* [172] and *Salmonella* Typhimurium, respectively. All these studies underscore the importance of small RNA in both antibiotic resistance as well as susceptibility, making them novel targets for drug development. To understand the role of small RNA in antibiotic resistance and susceptibility, I have divided the following three sections based on the strategies the multi-drug-resistant bacteria employ to curb the effects of the antibiotics.

### 4.1. Inhibiting Drug Uptake Systems

As detailed in the previous paragraphs, antibiotics enter bacterial cells through various mechanisms, ranging from simple diffusion to energy-dependent transport. To hinder antibiotic entry, certain bacteria employ sRNAs that target membrane proteins involved in antibiotic uptake, as discussed in the following section.

#### 4.1.1. GcvB

One of the key players of antibiotic resistance in *E. coli* and *Vibrio* is the small RNA GcvB, a 205-nt sRNA, transcribed divergently from the *gcvA*; the latter encodes for the transcriptional activator of the glycine cleavage (GCV) pathway. In the presence of glycine, transcription of *gcvB* is activated by GcvA and GcvR. Although GcvB was initially associated with the acid stress response [173], recent studies have elucidated its role in antibiotic resistance. Deleting *gcvB* in *E. coli* leads to heightened sensitivity to various aminoglycoside antibiotics, including neomycin, streptomycin, kanamycin, and kasugamycin, even at low concentrations [174]. GcvB consists of five stem-loops (SL1-5) with three conserved seed sequences, R1, R2, and R3, to regulate multiple target genes [175]. The G/U-rich R1 seed sequence located between SL1 and SL2 can base pair with the majority of target mRNAs. Although the seed sequence R2 primarily repress CycA mRNA in *E. coli* and *Salmonella*, whereas the R3 seed sequence, located in SL4, regulates several mRNAs, including global transcriptional regulators PhoP, Lrp, and SroC [176,177] (Figure 3a).

GcvB contributes to antibiotic resistance in *E. coli* by posttranscriptionally downregulating the expression of the inner membrane protein CycA. In addition to its role in amino acid (D-glycine, D-serine, D-alanine) import, CycA facilitates the uptake of aminoglycoside antibiotics such as streptomycin and gentamicin. GcvB mediates this regulation by base pairing with *cycA* mRNA, promoting its degradation and thereby reducing CycA protein levels, ultimately limiting the entry of aminoglycoside antibiotics into the cell [178]. Two regions of GcvB, spanning nucleotides +124 to +161 and +73 to +82, exhibit complementarity to the CycA mRNA. Transcriptional fusions of *cycA* to *lacZ* revealed that this CycA mRNA region is essential for GcvB-mediated regulation [179]. Muto et al. deciphered that either of the abovementioned complementary regions of GcvB can independently base pair with and repress *cycA-lacZ*, with significant repression loss occurring only when both regions are mutated [174]. Additionally, mutations in *cycA* can lead to increased resistance to aminoglycoside by reducing its uptake into the bacterial cell. Moreover, GcvB plays a role in promoting mutagenic break repair (MBR) by enabling a robust RpoS response while suppressing the competing SigE (membrane stress) response. *gcvB* deletion impairs MBR and RpoS-dependent transcription without reducing RpoS protein levels. Restoring RpoS levels or inhibiting SigE response induction rescues these defects, suggesting that GcvB maintains membrane integrity, resulting in decreased SigE activation. This balance prevents SigE from outcompeting RpoS for RNA polymerase, thereby supporting mutagenesis [180]. Furthermore, using transcriptomics- and proteomics-based approaches, Liu et al. identified the role of GcvB in regulating the type IIII secretion system in the marine bacterium *Vibrio alginolyticus*, thereby expanding its role in bacterial virulence [175].

#### 4.1.2. MicF

The micF RNA is a 93-nt sRNA that plays a significant role in the regulation of antibiotic resistance in *E. coli*. It primarily functions by binding to the mRNA of the outer membrane porin protein OmpF, preventing its translation [51,181]. OmpF forms trimeric channels that allow the passive diffusion of small hydrophilic molecules, including nutrients and antibiotics, across the outer membrane [182]. By repressing OmpF production, MicF reduces the permeability of the bacterial cell membrane to antibiotics, thereby decreasing the intracellular concentration of these drugs and enhancing bacterial survival. The expression of MicF is tightly regulated by several global stress response regulators, such as MarA, Rob, and SoxS [181] (Figure 3a). These regulators are activated under various stress conditions, including the presence of antibiotics, and oxidative stress conditions. When activated, they induce the expression of MicF, leading to a rapid decrease in OmpF levels and an increase in antibiotic resistance.

Carrier et al. recently elucidated the role of MicF in the activation of the outer membrane porin OppA [183]. OppA plays a key in the survival of uropathogenic *E. coli* by importing peptides, allowing the bacteria to thrive in the otherwise nutrient-limited and hostile urinary tract [137]. Additionally, OppA has also been implicated in antibiotic sensitivity in UPEC, particularly in response to polymyxin B [184]. The uptake of certain toxic compounds, including the synthetic tripeptide tri-L-ornithine [107] and the novel antibiotic GE81112 [108], is also dependent on OppA. Under standard growth conditions, GcvB and Hfq inhibit ribosomal binding to the SD sequence, resulting in low *oppA* expression. However, under membrane stress, MicF is activated and binds to the 5′ UTR of the OppA transcript, enhancing translation and leading to increased *oppA* expression.

#### 4.1.3. EsrA and Sr0161

*Pseudomonas aeruginosa* is a prominent opportunistic human pathogen responsible for approximately 3000 deaths annually in the United States [185]. Multi-drug-resistant (MDR) *P. aeruginosa* infections are often treated with carbapenem antibiotics like meropenem; however, 2–3% of clinical isolates produce carbapenemase, rendering them resistant to this antibiotic class. Carbapenem uptake in *P. aeruginosa* is mediated by the major porins OprD and OpdP, with *oprD* deletion increasing carbapenem resistance in MDR isolates [186].

Using Hi-GRIL-seq, Sr0161 and ErsA were identified as sRNAs that negatively regulate *oprD* expression via interactions with its 5′ UTR [187] (Figure 3a). The overexpression of Sr0161 enhances resistance to meropenem [187], while the deletion of *sr0161* or *ersA* significantly increases susceptibility to this antibiotic. Both *esrA* and *sr0161* are transcribed by stress-induced sigma factor 22. Sigma factor 22 (σ^22^), also known as AlgT or AlgU, in *Pseudomonas aeruginosa* is a key regulator of the alginate biosynthesis pathway and plays a central role in the transition to a mucoid phenotype during chronic infections. The RNA chaperone Hfq also contributes to carbapenem resistance by regulating OprD and OpdP porins, as shown by the increased carbapenem susceptibility observed in *hfq* deletion mutants [188].

### 4.2. Modification of Cell Envelope

Lipopolysaccharides (LPSs) are crucial components of the outer membrane of Gram-negative bacteria, playing key roles in maintaining membrane integrity, stress response, and virulence [189]. LPSs consist of three main parts: lipid A, a core oligosaccharide, and an O-antigen. The lipid A portion anchors LPS to the membrane and is a target for the polymyxins class of antibiotics such as colistin [190]. However, modifications to LPS, such as the addition of phosphoethanolamine, which changes the charge and structure of the lipid A, can reduce susceptibility to polymyxins. Polymyxin resistance in *E. coli* is mediated by the enzyme EptB, which modifies LPS with phosphoethanolamine [184]. The translation of *eptB* mRNA is inhibited by the sRNA MgrR, which is itself suppressed by the sponge sRNA SroC. As a result, the loss of MgrR enhances, while the loss of SroC reduces resistance to polymyxin B. Additionally, *eptB* expression is negatively regulated by the sRNA ArcZ, whose levels are governed by the ArcA–ArcB two-component system, which senses aerobic and anaerobic conditions [191]. Interestingly, the deletion of Hfq, a protein essential for the activity of these sRNAs, results in increased susceptibility to polymyxin B. This phenomenon may be attributed to a compromised cell envelope stress response mediated by the sigma factor RpoE. RpoE not only promotes *eptB* transcription but also regulates other Hfq-dependent sRNAs involved in LPS biogenesis and modification. The following section discusses the role of two small RNAs that provide antibiotic resistance by modification of the bacterial cell membrane.

#### 4.2.1. SprX 

*Staphylococcus aureus* is a major cause of bacteremia, infective endocarditis, and osteomyelitis, and frequently colonizes medical implants [192]. The emergence of MRSA strains has severely limited treatment options, often leaving glycopeptide antimicrobials, which target peptidoglycan cell wall synthesis, as the primary therapeutic option [193]. The sRNA SprX encoded within a pathogenicity island is highly conserved among *Staphylococcus aureus* strains and is present in several *S. aureus* phages [194,195,196,197]. SprX controls virulence factors such as delta-haemolysin and clumping factor B by interacting with *hld* and *clfB* transcripts, while also influencing IsaA expression and biofilm formation [195]. In *S. aureus* HG001, SprX plays a role in modulating glycopeptide resistance and complement-binding protein [196,197]. Most strains carry a single copy, while the Newman strain has three copies with minor nucleotide differences. SprX plays a pivotal role in modulating antibiotic resistance mechanisms by directly base pairing with the translation initiation site of SpoVG mRNA, a key regulator of capsule production, virulence, and cell wall metabolism. This interaction prevents ribosome loading, thereby suppressing SpoVG protein expression [198]. SpoVG regulates critical pathways associated with methicillin and glycopeptide resistance, including the activation of the *lytSR* operon, the glycine glycyltransferases (*femAB*), which mediate peptidoglycan crosslinking and repression of the murein hydrolase *lytN*. By inhibiting SpoVG, SprX increases MRSA susceptibility to glycopeptides. 

#### 4.2.2. Sr006

The sRNA Sr006, transcribed from the intergenic region between 182,570 and 182,693 bp on the minus strand, was identified in chimeric RNA molecules with lipid A diacylation enzyme PagL mRNA [199] and functions as a positive posttranscriptional regulator of PagL in *P. aeruginosa* virulence and host adaptation, as revealed by Hi-GRIL-seq. Western blot and RT-qPCR analyses demonstrated that the overexpression of Sr006 significantly increased PagL mRNA and PagL protein levels, while the deletion of Sr006 had no effect. Functional studies revealed that the Sr006-mediated upregulation of PagL resulted in enhanced deacylation of lipid A, shifting it to a less pro-inflammatory form, as confirmed by MALDI-MS and GC-FID analyses. Furthermore, Sr006-dependent regulation of PagL increased polymyxin B resistance in *P. aeruginosa* strains lacking LPS aminoarabinose modifications. Unlike other sRNA-mediated regulatory mechanisms requiring the RNA chaperone Hfq, Sr006’s regulation of PagL was Hfq-independent, suggesting an alternative mechanism of action. This regulatory pathway may enable *P. aeruginosa* to adapt to the host environment, potentially through modulation of Sr006 expression in response to host interactions.

### 4.3. Activation of Efflux Pumps

The tripartite efflux pump systems in *E. coli*—AcrAB-TolC, MacAB-TolC, MdtABC-TolC, and EmrAB-TolC—play crucial roles in multi-drug resistance by expelling a variety of toxic compounds [200] (Figure 3a). The AcrAB-TolC system consists of AcrA, a periplasmic adapter protein that bridges the inner and outer membranes, AcrB, a Resistance-Nodulation-Division (RND) transporter responsible for recognizing diverse substrates such as antibiotics and bile salts, and TolC, an outer membrane channel [201]. The *acrAB* genes are organized within an operon, while *tolC* is located separately on the *E. coli* chromosome within an operon that includes *ygiB* and *ygiC*. Both the *acrAB* and *tolC* promoters are regulated by the MarA, SoxS, and Rob transcriptional regulators and are upregulated in response to superoxide stress and exposure to heavy metals, bile salts, and antibiotics, reflecting the efflux pump’s role in exporting these harmful substances from the cell [202] (Figure 3a). In addition to AcrAB-TolC efflux pump, TolC was found to be associated with MacA-MacB and MtdA-MdtB-MdtC to form MacAB-TolC [200,203] and MdtABC-TolC efflux pumps, respectively [204]. The MacAB-TolC pump exports macrolides like erythromycin and secretes virulence factors, whereas the MdtABC-TolC system confers resistance to bile salts and heavy metals, aiding bacterial colonization in the gut. Lastly, the EmrAB-TolC system, part of the Major Facilitator Superfamily (MFS), uses a proton/drug antiporter mechanism to resist hydrophobic antibiotics [205] (Figure 3a). Collectively, these efflux pumps are vital for bacterial survival in hostile environments and contribute significantly to antibiotic resistance. Recent works have deciphered the regulations of these multi-drug efflux pumps by sRNAs, as discussed in the following paragraphs.

#### 4.3.1. CsiR

Strong evidence indicates that antibiotics are a major driving force behind bacterial resistance [206]. Ciprofloxacin, a third-generation synthetic quinolone, is widely used in clinical and agricultural settings, exerting its antibacterial effects by inhibiting bacterial DNA gyrase and topoisomerase IV and effectively targeting *P. aeruginosa*, *E. coli*, *Proteus vulgaris*, and other Enterobacteriaceae [207]. CsiR, a sRNA, functions as a negative regulator of *emrB*, a multi-drug efflux pump gene that plays a crucial role in ciprofloxacin resistance in *P. vulgaris* strain P3M. Gene expression analysis revealed that EmrB was significantly upregulated under ciprofloxacin treatment. Interestingly EmrB expression was even higher in a Δ*csiR* mutant strain, confirming that CsiR inhibits EmrB expression, thereby decreasing ciprofloxacin efflux and reducing bacterial resistance. Bioinformatics predictions identified a specific binding site for CsiR in 5′ UTR, near the SD sequence of EmrB mRNA, suggesting a direct posttranscriptional regulatory interaction. This was experimentally validated through microscale thermophoresis (MST) assays, which showed a strong binding affinity between wild-type CsiR and EmrB mRNA, with a dissociation constant of 11.5 ± 3.56 μM [208]. Mutation of the predicted EmrB binding site in CsiR resulted in abolishing its interaction with EmrB. Furthermore, in vivo studies confirmed that mutating this binding site in CsiR led to a loss of repression, restoring EmrB expression to levels observed in the Δ*csiR* strain and significantly increasing bacterial survival in the presence of ciprofloxacin. These findings strongly suggest that the CsiR-mediated repression of EmrB represents a key regulatory mechanism controlling antibiotic resistance in *P. vulgaris*, highlighting the potential of targeting sRNA–efflux pump interactions for antimicrobial therapy [208]. The absence of CsiR also affected resistance to polymyxin B and erythromycin, indicating its multifunctional role in antibiotic resistance.

#### 4.3.2. PA0805.1

*Pseudomonas aeruginosa* initiates swarming motility in response to semisolid surfaces and nitrogen sources relevant to the human lung. Although the transcriptional regulation of swarming has been well studied, posttranscriptional regulation by sRNAs remains underexplored. Coleman et al. found that the sRNA PA0805.1 was upregulated 5-fold under swarming conditions. The overexpression of PA0805.1 in wild-type (WT) PAO1 led to reduced swarming, swimming, and twitching motilities, along with increased adherence, cytotoxicity, and tobramycin resistance, while its deletion (Δ*PA0805.1*) increased tobramycin susceptibility [209].

Transcriptomic and proteomic analyses of the PA0805.1-overexpressing strain revealed significant alterations in gene and protein expression, with 1,121 differentially expressed genes and 258 proteins identified. Among these were 106 transcriptional regulators, two-component systems, and sigma/anti-sigma factors implicated in antimicrobial resistance and virulence. Notably, the upregulation of multi-drug efflux systems *mexXY* and *mexGHI-opmD* provides a mechanistic basis for increased tobramycin resistance (Figure 3a). Additionally, ClpD, a protease essential for biofilm dispersal and virulence, was upregulated, potentially contributing to adaptive resistance. Further regulatory changes included the downregulation of type IV pilus genes and transcription factors such as PilGH and ExsD, the latter being a negative regulator of the type III secretion system (T3SS). In contrast, key regulators involved in virulence, quorum sensing, and biofilm formation—AlgR, LasR, RhlR, MvaT, and PsrA were differentially expressed, reinforcing a shift towards an antibiotic-resistant and persistent phenotype [209]. These findings underscore the role of PA0805.1 in modulating antibiotic resistance pathways in *P. aeruginosa*, particularly through efflux system activation and biofilm-associated mechanisms, highlighting its potential as a target for therapeutic intervention.

#### 4.3.3. SdsR and RyeA

Small RNAs (sRNAs) SdsR and RyeA in *E. coli*, transcribed from opposite strands of the same genomic locus, exemplify a unique sRNA-mediated toxin–antitoxin (T/A) system [210]. The ectopic expression of SdsR during exponential growth caused Hfq-dependent cell death and filamentation, with RNA-seq analysis revealing the altered expression of 209 genes, including the repression of *yhcB*, encoding an inner membrane protein critical for survival. The overexpression of RyeA mitigated SdsR-induced toxicity, highlighting an antagonistic interplay between the two sRNAs. These findings uncover a novel regulatory role for SdsR in bacterial stress responses and emphasize its functional relationship with RyeA in maintaining cellular homeostasis [210].

Importantly, SdsR is essential for the posttranscriptional regulation of the AcrAB-TolC efflux system, directly base pairing upstream of the TolC ribosomal binding site to repress its expression, thereby reducing TolC protein production [211]. This repression leads to increased susceptibility to antibiotics such as novobiocin, indicating that SdsR negatively impacts the bacterium’s ability to expel toxic compounds. Similarly, the overexpression of SdsR has been shown to heighten susceptibility to quinolone antibiotics, including levofloxacin, nalidixic acid, and norfloxacin, in multi-drug-resistant strains of *E. coli* and *Salmonella*.

Additionally, SdsR has been linked to broader stress responses and genetic regulation. For instance, SdsR appears to influence bacterial mutagenesis, as its interaction with *mutS* has been associated with an increased mutation rate under subinhibitory ampicillin treatment [212]. Furthermore, SdsR overexpression was found to reduce biofilm formation in *E. coli*, suggesting that it may influence bacterial surface attachment and persistence. These findings collectively highlight SdsR as a key modulator of antibiotic susceptibility and stress adaptation, though its precise physiological roles warrant further investigation.

### 4.4. Biofilm-Mediated Increased Drug Resistance in Bacterial Pathogens

One of the adaptive strategies employed by bacteria to evade the host immune response and antibiotic action is biofilm formation [213,214,215]. Biofilms are structured communities of bacteria, encased in an extracellular polymeric substance (EPS) composed of polysaccharides, proteins, lipids, and extracellular DNA [216]. This EPS matrix acts as a physical barrier, limiting antibiotic penetration and creating chemical gradients that reduce the effectiveness of antibiotics [217]. The formation of biofilms is tightly regulated by transcriptional factors. In *P. aeruginosa*, key regulators include AlgR, which promotes the production of alginate (a major biofilm polysaccharide), and the two-component systems GacS/GacA and BfiSR, which regulate biofilm maturation [218]. The quorum sensing regulators LasR and RhlR also play crucial roles in coordinating biofilm development by inducing EPS production and virulence factor expression [219].

sRNAs are now recognized for their crucial role in biofilm formation, a prevalent multicellular bacterial lifestyle across various bacterial species [41,220]. In *E. coli*, CsrB and CsrC sequester CsrA, resulting in the increased expression of the genes involved in extracellular polysaccharide synthesis. Similarly, in *S. enterica*, small RNAs ArcZ and SdsR regulate the transcriptional switch CsgD, which governs biofilm matrix production and motility repression. Additional sRNAs, McaS, RprA, and OmrA/OmrB, integrate environmental signals to modulate *csgD* expression, thereby influencing lifestyle transitions. The second messenger c-di-GMP further reinforces biofilm formation by promoting extracellular polysaccharide production [221]. Moreover, in *V. cholerae*, the sRNA RybB represses *ompA*, modulating membrane permeability to favor biofilm formation and environmental adaptation [108]. Finally, in *Bacillus subtilis*, BsrG and SR1 contribute to biofilm formation, with SR1 modulating metabolic pathways for extracellular polysaccharide synthesis and BsrG enhancing stress adaptation. Collectively, these regulatory sRNAs enable bacteria to fine-tune biofilm formation in response to environmental cues.

Although sRNAs play a crucial role in regulating biofilm formation across multiple bacterial pathogens, this review focuses on three specific sRNAs involved in biofilm formation in *Pseudomonas* sp. The recent rise in multi-drug-resistant (MDR) *P. aeruginosa* has become a significant global health concern, largely driven by the overuse and misuse of antibiotics in clinical and environmental settings. Biofilm formation is a key contributor to this multi-drug-resistant phenotype, enhancing bacterial survival and persistence. A deeper understanding of the molecular mechanisms underlying biofilm formation should provide new avenues for the development of targeted therapeutic strategies.

#### 4.4.1. AS1974

The 127-nt sRNA AS1974 has been implicated in biofilm-associated antibiotic resistance in *Pseudomonas* sp. Hfq RIP-seq analysis identified AS1974 as being downregulated in MDR *P. aeruginosa* isolates, with its overexpression sensitizing cells to aminoglycosides, cephalosporins, and carbapenems [222,223], suggesting AS1974 as a negative regulator of antimicrobial resistance. This is achieved by AS1974-mediated repression of *ndvB*, a gene encoding a biofilm-associated protein required for the synthesis of periplasmic glucans that sequester antibiotics and prevent their entry into the bacterial cell. The deletion of *ndvB* sensitizes biofilms to gentamicin and ciprofloxacin [223]. Transcriptomic analysis revealed that AS1974 sRNA plays a regulatory role in multiple drug resistance pathways by downregulating key resistance genes, including *mexD*, a component of multi-drug efflux system MexC-MexD-OprJ, *prc*, periplasmic tail specific protease, an enzyme involved in the processing of C-terminal region of penicillin binding protein 3, and *chtA*, a TonB-dependent siderophore receptor (Figure 3b).

To investigate the regulation of *AS1974*, researchers analyzed its promoter region in drug-susceptible and multi-drug-resistant (MDR) strains. Chromatin immunoprecipitation followed by qPCR (ChIP-qPCR) revealed stronger RNA polymerase binding to the *AS1974* promoter in susceptible strains, hinting toward a possible epigenetic regulation of AS1974 that downregulates its transcription in resistant strains akin to the methylation-mediated regulation of P-fimbriae (discussed in Section 2.1.1). Bisulfite genomic sequencing identified three methylation sites upstream of the *AS1974* promoter in resistant strains, which are missing in susceptible strains. These sites were located at −16, −66, and −73 relative to the transcriptional start site (TSS) of AS1974 (Figure 3b). A GFP reporter assay demonstrated that *AS1974* transcription was suppressed in resistant strains, confirming a functional impact of methylation. These results suggest that promoter methylation inhibits *AS1974* transcription by restricting RNA polymerase binding, contributing to drug resistance in MDR strains [223].

#### 4.4.2. PhoS

PhoS is a 66-nt sRNA that is encoded at the 3′UTR of *phoR*, although the expression of PhoS is driven by its own promoter (Figure 3c). PhoR is a sensor kinase that regulates bacterial responses to phosphate availability by toggling between kinase and phosphatase modes. Under phosphate-limiting conditions, PhoR functions as a kinase, phosphorylating its cognate response regulators, PhoB or PhoP, which then activate the expression of genes involved in the adaptation to phosphate scarcity. Conversely, under phosphate-replete conditions, PhoR assumes a phosphatase role, maintaining PhoB/PhoP in an inactive, dephosphorylated state (Figure 3c).

Studies on *E. coli* and *B. subtilis* have identified two mechanisms regulating PhoR activity. In *E. coli*, the predominant pathway involves the PstSCAB/PhoU complex, which transduces phosphate transporter conformational changes to PhoR. However, in *B. subtilis*, an alternative mechanism links PhoR regulation to wall teichoic acid metabolism, where biosynthetic intermediates modulate PhoR autokinase activity. PhoS regulates biofilm formation in *Bacillus* sp. and is induced under phosphate limitation [224]. The disruption of *phoS* reduced EPS production, whereas the overexpression increased EPS levels, confirming its role in biofilm synthesis. PhoS enhances the translation of PhoP mRNA by preventing an inhibitory structure at its 5′UTR, forming a self-reinforcing autoregulatory loop that allows *Bacillus* sp. to adapt to phosphate scarcity. Through PhoP, PhoS promotes biofilm formation by influencing the expression of operons involved in cell wall metabolism, such as *tuaA-H*, which facilitates teichuronic acid synthesis. Additionally, PhoS potentially affects biofilm development through the PhoP-regulated *comQXPA* operon, impacting quorum sensing and surfactin signaling. While the deletion of *phoS* weakly reduces biofilm formation, its overexpression strongly enhances it, likely due to the regulatory loop amplifying PhoP levels. Transcriptomic analyses reveal that PhoS modulates key biofilm-related genes, though it does not directly target *eps* or *tapA-sipW-tasA* operons in *B. subtilis* and *B. velezensis* strain FZB42.

#### 4.4.3. SrbA

The small RNA SrbA plays a crucial role in *P. aeruginosa* biofilm formation. A *srbA* deletion mutant exhibited a 66% reduction in biofilm formation while maintaining normal surface attachment in a *C. elegans* slow-killing model. The complementation of *srbA* restored wild-type biofilm levels, indicating SrbA’s essential function. Genome-wide analysis (using TargetRNA2 and BLASTn) identified 61 putative mRNA targets of SrbA, including genes linked to biofilm formation, motility, metabolism, and stress response, suggesting broad posttranscriptional regulation. RT-qPCR revealed significant transcriptomic changes in the *srbA* mutant during biofilm growth, with 12 genes downregulated ≥4-fold and 7 upregulated ≥2-fold. These findings suggest that SrbA modulates biofilm formation by influencing mRNA stability and translation efficiency, potentially via interactions with the ribosome binding site or RNase E recruitment [225].

As illustrated above, sRNAs contribute to antibiotic resistance in bacterial pathogens via diverse mechanisms. Notably, these regulatory RNAs also play critical roles in quorum sensing pathways, which can further potentiate multidrug resistance, as explored in the following section.

## 5. Role of sRNA in Quorum Sensing

Bacteria regulate population density through a process known as quorum sensing (QS), which involves the production, release, accumulation, and detection of extracellular signaling molecules termed autoinducers (AIs) [15,17,226]. Quorum sensing enables bacterial communities to coordinate responses to chemical signals, promoting cooperative behaviors that enhance fitness [17,227]. These include coordinated virulence, evading host immune responses, and establishing infections [228]. This mechanism is present in both Gram-positive and Gram-negative pathogens, though the signaling molecules triggering QS circuits differ between them. Gram-positive bacteria typically respond to processed oligopeptides [56], whereas Gram-negative bacteria respond to homoserine lactones as autoinducers [17]. Previous studies on *P. aeruginosa* [229], *S. aureus* [230], and *V. cholerae* [231] reveal that these pathogens use QS to regulate the production of virulence factors, including toxins, proteases, siderophores, and biofilm formation, which enhance their survival within the host. QS also facilitates immune evasion by regulating the expression of immune-modulatory molecules, allowing bacteria to remain undetected until they reach a critical population density. Additionally, QS-mediated biofilm formation protects bacteria from antibiotics and immune clearance, contributing to chronic infections [52,213]. Emerging evidence suggests that host organisms can interfere with QS through quorum quenching, a strategy in which the host degrades or inhibits bacterial signaling molecules to prevent infection. Understanding QS in host–pathogen interactions is essential for developing novel anti-virulence therapies that target bacterial communication without promoting antibiotic resistance.

This section will explore the mechanism of quorum sensing and its role in bacterial pathogenesis and host–pathogen interactions, focusing on two Gram-negative representatives, *Vibrio* sp. and *P. aeruginosa*, and one Gram-positive representative, *S. aureus*.

### 5.1. Qrr 1-5

Quorum sensing (QS) in *V. harveyi* and *V. cholerae* is controlled by sRNAs called quorum regulatory RNAs (Qrrs) and the RNA chaperone Hfq. This system enables bacteria to synchronize group behaviors, including bioluminescence, virulence factors production, type VI secretion, and biofilm formation as a function of cell density. *V. harveyi* produces three autoinducers—an acyl homoserine lactone-HAI-1 (also known as harveyi auto inducer 1), a furanosyl borate diester-AI-2 (also known as **autoinducer-2**), and (*S*)-3-hydroxytridecan-4-one-CAI-1 (also known as cholerae autoinducer 1)—synthesized by LuxM, LuxS, and CqsA, respectively [232,233,234]. These autoinducers are detected by membrane-bound two-component sensors, LuxN, LuxQ, and CqsS (Figure 4). The autoinducers regulate QS through a phosphorelay system. At low cell density, when autoinducer concentrations are minimal, these sensors function as kinases, phosphorylating LuxU, which subsequently phosphorylates LuxO. Phosphorylated LuxO (LuxO–P), in conjunction with σ⁵⁴, activates the transcription of five Qrr sRNAs (Qrr1–5) [233]. These Qrr sRNAs bind to Hfq and interact with target mRNAs through specific base pairing to regulate gene expression via catalytic degradation, coupled degradation, sequestration, or activation-induced degradation. Each Qrr sRNA contains four stem-loops, with the first and second being critical for base pairing and stability. This regulatory network represses the master QS regulator LuxR, preventing QS-induced gene expression at low cell density. At high cell density, accumulated autoinducers switch the sensors from kinases to phosphatases, resulting in LuxO dephosphorylation and the subsequent cessation of Qrr transcription. This permits LuxR translation and the activation of QS-regulated genes, leading to behaviors such as bioluminescence and biofilm formation (Figure 4).

In *V. cholerae*, the QS system operates similarly but lacks the HAI-1 pathway, relying on AI-2 and CAI-1 autoinducers [235,236]. The phosphorelay system involving LuxU and LuxO regulates QS gene expression. However, unlike *V. harveyi*, *V. cholerae* produces four Qrr sRNAs (Qrr1–4), which redundantly repress HapR, the homolog of LuxR. Additionally, the VarS–VarA two-component sensory system modulates QS by regulating the activity of the CsrA protein, which indirectly influences LuxO activity [237]. At low cell density, CsrA enhances LuxO–P activity, promoting Qrr-mediated repression of HapR. At high cell density, VarS–VarA inhibits CsrA, allowing HapR expression, which suppresses virulence factor production and biofilm formation.

A key player of *Vibrio cholerae* virulence is the type VI secretion system (T6SS). T6SS is a QS-regulated, contact-dependent protein delivery apparatus that can eliminate competing cells [238,239]. Structurally, T6SS assembles into a membrane-spanning, spear-like device loaded with toxic effector proteins [240]. Upon deployment, the effectors penetrate competitor cell walls. To prevent self-toxicity, T6SS-active cells produce immunity proteins that neutralize the toxic effectors [241]. Additional protective mechanisms, such as exopolysaccharide and capsular polysaccharide production, also safeguard against incoming T6SS-mediated attacks [242]. The Qrr sRNAs repress *t6ss* expression via two mechanisms: direct repression of the primary *t6ss* gene cluster by interacting with their mRNA, and indirect repression by inhibiting HapR, the HCD (high cell density) activator of auxiliary *t6ss* gene clusters [238]. Since T6SS toxin deployment requires direct cell-to-cell contact, limiting maximal T6SS machinery production to HCD may enhance killing efficiency and reduce energy expenditure. Interestingly, Huber et al. identified an RNA sponge, QrrX, which base pairs with and inactivates the Qrr1-4 sRNAs [243]. The transcription of *qrrX* is driven by QrrT, a previously uncharacterized LysR-type transcriptional regulator. Findings from this study suggest that QrrX and QrrT play essential roles in facilitating the transition from individual to community behaviors in *V. cholerae*.

### 5.2. PhrS and ReaL

*P. aeruginosa* is an opportunistic pathogen that causes severe infections in immunocompromised individuals and cystic fibrosis patients [244]. In *P. aeruginosa*, four interconnected QS systems—Las, Rhl, Pqs, and Iqs—regulate over 300 virulence-related genes [229,245]. The Las and Rhl systems use N-acyl homoserine lactone (AHL) signals, with *LasI* producing 3-oxo-C12-HSL to activate *LasR*, leading to the expression of virulence factors like elastases and exotoxins. The Rhl system, with RhlI producing C4-HSL, interacts with RhlR to regulate factors such as rhamnolipid and hydrogen cyanide. The Pqs system synthesizes quinolones (PQS signals) via PqsABCD, activating PqsR. PQS acts as a critical link between the Las and Rhl systems, regulating genes involved in biofilm formation, iron acquisition, and the production of the redox-active toxin pyocyanin (PYO) [246,247]. PYO contributes to pathogenicity by generating ROS, inducing oxidative stress, and damaging host tissues, while also exhibiting antimicrobial activity against competing microorganisms [248]. The Iqs system, with *ambBCDE*, produces IQS signals. These QS systems function hierarchically, with Las at the top, regulating Rhl and Pqs, while Pqs also modulates Las and Rhl.

The sRNA PhrS plays a central role in modulating QS and virulence in *P. aeruginosa* [249]. Its expression is controlled by the oxygen-responsive transcription factor ANR and is enhanced during hypoxia or the stationary phase. A putative ANR binding site upstream of the *phrS* promoter has been identified, and *phrS* expression is abolished in ANR-deficient mutants [249]. Additionally, Hfq indirectly regulates *phrS* by stabilizing ANR. PhrS promotes the translation of *pqsR*, a key activator of PQS synthesis through structural rearrangements in hypoxic conditions, ultimately enhancing PQS and PYO production. PYO contributes to immune evasion through ROS generation and the modulation of IL-8-driven immune responses. PhrS-deficient strains exhibit impaired PYO production, highlighting its critical role in *P. aeruginosa* virulence. In addition, PhrS downregulates the expression of cytochrome bo3 oxidase genes (*cyoAB*), facilitating a shift in oxygen utilization pathways under low oxygen [249]. By overcoming the oxygen-dependent suppression of PQS, PhrS supports QS-dependent gene expression under hypoxia, a condition common in biofilms and cystic fibrosis lung infections. Furthermore, PhrS-mediated PQS synthesis promotes extracellular DNA release, a crucial component for biofilm stability. These findings suggest that targeting PhrS may offer a novel strategy to mitigate *P. aeruginosa* infections.

In addition to PhrS, the sRNA ReaL in *Pseudomonas aeruginosa* plays a role in virulence by linking the Las and Pqs quorum sensing (QS) systems [250]. The deletion of *reaL* reduces virulence in the *Galleria mellonella* infection model, while its overexpression leads to a hyper-virulent phenotype. ReaL is negatively regulated by the Las QS regulator LasR and enhances the synthesis of the pqs quinolone signal PQS by posttranscriptionally activating PqsC. The region from −13 to +32 of PqsC mRNA is predicted to form a stable stem-loop structure, sequestering the SD sequence and the AUG start codon through base pairing with the RII region, effectively acting as an anti-SD/AUG element and repressing PqsC translation. When ReaL binds to the RI-RII region of PqsC, it induces a conformational change that extrudes a loop containing the SD and AUG sequences. This structural rearrangement facilitates the recruitment of the 30S ribosomal subunit to the mRNA, enabling translation initiation [250]. This affects key virulence traits such as PYO production, biofilm formation, and swarming motility. Additionally, ReaL responds to host-relevant conditions like temperature and oxygen levels and is upregulated in the stationary phase by the alternative σ factor RpoS. These regulatory mechanisms fine-tune PQS synthesis and contribute to *P. aeruginosa* virulence.

### 5.3. AmiL and PqsS

In addition to PhrS and ReaL discussed above, recent studies have identified AmiL, a QS-regulatory sRNA that modulates virulence in *P. aeruginosa* by targeting genes involved in key virulence factors and phenotypes such as PYO, elastase, rhamnolipid, biofilm formation, swarming motility, hemolysin, and cytotoxicity [251]. However, unlike AmiL and PqsS, which positively regulate PYO, AmiL acts as a negative regulator. The 100-nt sRNA AmiL is a leader of the *amiEBCRS* operon involved in biofilm formation [252]. Using transcriptomics, Pu et al. revealed that AmiL expression is negatively regulated by RhlR [251]. As discussed above, PhrS directly targets PhzC (a member of the operon responsible for the synthesis of PYO) and regulates LasI and RhlI expression [249,253]. This defines a new las/rhl (RhlR)-AmiL-PhzC/Las/Rhl signaling cascade, emphasizing AmiL’s central role in PAO1 virulence. While AmiL downregulates lasI during early growth, it upregulates rhlI and C4-HSL production in later stages [251]. It also promotes swarming motility by enhancing rhamnolipid production. This dual role in las and rhl regulation highlights AmiL’s function in fine-tuning *P. aeruginosa*’s QS-sRNA network to balance virulence.

Another Hfq-dependent small RNA PqsS was recently discovered in *P. aeruginosa* PAO1, which enhances bacterial pathogenicity by regulating the *Pseudomonas* quinolone signal (PQS) quorum sensing (pqs QS) system [254]. PqsS promotes acute infections by inducing host cell death and stimulating rhamnolipid-mediated swarming motility, while simultaneously reducing chronic infection traits such as biofilm formation and antibiotic resistance. Mechanistically, PqsS represses *pqsL* transcript, leading to increased PQS levels and reinforcing Pqs QS [254]. Additionally, a PQS-rich environment upregulates PqsS expression, establishing a positive feedback loop. Further analysis revealed that PqsS interacts with PqsL mRNA, recruiting RNase E to facilitate its degradation. These findings provide valuable insights into *P. aeruginosa* pathogenesis and offer potential avenues for targeted therapeutic strategies

### 5.4. SprD

The Gram-positive pathogen *S. aureus* utilizes quorum sensing (QS) to regulate virulence and biofilm formation, primarily through the Agr (Accessory gene regulator) system. The Agr system operates via a two-component signaling mechanism involving the autoinducing peptide (AIP) and the AgrC-AgrA regulatory circuit. AIP is synthesized as a precursor and processed into its active form, which binds to the membrane-bound sensor kinase AgrC, triggering a phosphorylation cascade that activates the response regulator AgrA. This activation leads to the expression of RNAIII, a regulatory RNA that controls virulence gene expression, including toxins and exoenzymes, while repressing surface proteins associated with early biofilm formation [98].

In addition to RNAIII, which was discussed in Section 2.2.2, another small RNA involved in QS in *Staphylococcus* is the small pathogenicity island RNA D (SprD) [255]. SprD is crucial for the virulence of *Staphylococcus aureus*, as its deletion in a murine sepsis model resulted in 100% survival by day 21, compared to complete mortality in mice infected with the wild-type strain. Complementing the mutant with *sprD* partially restored virulence (50% survival). Moreover, mice infected with the *ΔsprD* strain had smaller, uniformly discolored kidneys, whereas wild-type infections caused swollen, abscess-filled kidneys [255]. Bacterial counts revealed reduced bacterial persistence in the absence of *sprD*. While SprD regulates the immune evasion protein Sbi, altering Sbi expression alone did not affect virulence, suggesting that SprD targets additional factors essential for pathogenesis. The regulation of *sbi* mRNA by SprD in *S. aureus* involves direct antisense pairing, as shown by in silico predictions and confirmed through gel retardation assays. SprD binds a specific region of Sbi mRNA near its SD sequence and AUG initiation codon, forming a stable duplex that prevents ribosome loading and translation initiation [255]. Deletion mutants of both SprD and Sbi mRNA abolished this interaction, confirming the necessity of these binding regions.

Taken together, small regulatory RNAs are key regulators of quorum sensing in pathogens like *E. coli*, *Vibrio* spp., *P. aeruginosa*, and *S. aureus*. By modulating quorum sensing components, sRNAs influence bacterial communication, virulence, and biofilm formation, aiding adaptation and immune evasion. Their role in host–pathogen interactions make them promising targets for antimicrobial strategies to disrupt quorum sensing-mediated virulence.

## 6. Discussion

The study of bacterial sRNAs has greatly advanced due to the development of high-throughput sequencing technologies, allowing for a more comprehensive understanding of sRNA networks and their regulatory roles. In *Escherichia coli*, RNA sequencing (RNA-seq) has identified approximately 108 sRNAs, while sRNA interactome profiling has revealed between 1600 and 1900 sRNA-mRNA interactions. These extensive regulatory networks parallel the complexity of transcription factor interactions, which encompass ~300 transcription factors [256]. Crosslinking immunoprecipitation (CLIP) techniques, such as Hfq-CLIP and Rbp-CLIP, were developed that provide insights into sRNA–protein interactions by capturing RNA-binding protein-associated sRNAs, helping to define their functional partners. Additionally, CRISPR-based approaches, including CRISPR interference (CRISPRi) and CRISPR-Cas knockout systems, facilitate functional studies by selectively repressing or deleting sRNAs, thereby elucidating their regulatory roles in bacterial physiology. Similarly to transcription factors, sRNAs often function as regulatory “hubs”, influencing a broad array of target RNAs that collectively shape phenotypic outcomes. These small RNAs possess both conserved and species-specific features, which collectively enhance the competitive fitness of bacterial pathogens in their respective ecological niches (Appendix A). The development of sophisticated sequencing-based methodologies such as RIL-seq (RNA Interaction by Ligation and Sequencing) [257], GRIL-seq (Global RNA Interaction by Ligation and Sequencing) [258], Hi-GRIL-seq (High-Throughput GRIL-seq) [187,259] has revolutionized our ability to map these complex interactions. These methodologies provide a comprehensive framework for understanding sRNA function in bacterial stress responses, antibiotic resistance, and virulence. RIL-seq, for instance, captures transcriptome-wide RNA-RNA interactions mediated by RNA-binding proteins (RBPs) such as Hfq, revealing complex sRNA-mRNA interaction networks. GRIL-seq, on the other hand, specifically focuses on identifying sRNA interactions without relying on RBP immunoprecipitation, using in vivo proximity ligation to directly capture RNA duplexes. Meanwhile, Hfq-RIP-seq enables the genome-wide identification of Hfq-associated RNAs, further illuminating the role of this global RNA chaperone in posttranscriptional regulation. Together, these methodologies have significantly expanded our understanding of bacterial sRNA networks and their regulatory influence over critical cellular processes.

Recent advances in bacterial stress response and pathogenesis research have unveiled sophisticated regulatory networks that enable bacteria to survive harsh conditions, including extreme temperatures, oxidative stress, nutrient deprivation, and antibiotic exposure [260,261]. For example, the role of phase separation in stress granule formation has emerged as a critical mechanism for bacterial survival under stress [262]. Advances in multi-omics approaches, such as single-cell RNA sequencing and proteomics, have provided deeper insights into stress-induced regulatory circuits. Furthermore, CRISPR-based tools were employed to dissect stress-related genetic pathways, revealing new targets for antimicrobial development [263]. These findings not only expand our understanding of bacterial resilience but also offer potential strategies for combating antibiotic resistance and improving industrial microbial applications.

Similarly, emerging trends in QS research have unveiled novel strategies to modulate bacterial communication, offering potential therapeutic avenues against pathogenicity [15]. Additionally, the integration of quorum sensing inhibitors (QSIs) with traditional antibiotics has shown synergistic effects and enhanced antibacterial efficacy, providing new strategies for the clinical treatment of bacterial infections [264]. Furthermore, the application of exogenous QS signaling molecules, such as acyl-homoserine lactones (AHLs), was explored to enhance electrochemical activities in bioelectrochemical systems, indicating QS’s potential in bioengineering applications [265]. Collectively, these advancements underscore the pivotal role of QS in bacterial behavior and its promising exploitation in medical and industrial contexts.

Ongoing advancements highlight the crucial role of sRNAs in bacterial adaptation to antibiotics by regulating resistance and tolerance mechanisms. Exposure to sub-MICs of antibiotics induces broad transcriptional and proteomic changes, including alterations in sRNA expression profiles [266]. This adaptive response enables bacteria to evade growth inhibition and develop antibiotic tolerance. Specific sRNAs have been implicated in antibiotic resistance across multiple species. In *Salmonella* Typhimurium, for example, tigecycline treatment induces the sRNA SroA [267], whose deletion increases susceptibility to the drug, while ectopic expression restores resistance. Similarly, in *Clostridia*, exposure to clindamycin induces an sRNA located upstream of an ABC transporter gene, conferring resistance in *Staphylococcus* species [268]. Transcriptomic analyses of antibiotic-resistant pathogens, including Mtb, MRSA, and *P. putida*, have demonstrated that different antibiotics elicit unique sRNA expression profiles [269,270,271]. Some antibiotics induce extensive sRNA regulatory changes, while others influence only a limited subset of sRNAs. In *E. coli*, antibiotic stress triggers the activation of the alternative sigma factor RpoS through Hfq-dependent sRNAs such as RprA, which governs a range of stress response genes [272]. RpoS activation facilitates mutagenesis by upregulating error-prone DNA polymerase IV [273] and downregulating the DNA repair protein MutS via the sRNA SdsR. These processes enhance genetic diversity, accelerating the evolution of antibiotic resistance. Furthermore, RpoS and its associated sRNAs contribute to horizontal gene transfer by promoting plasmid conjugation, a key mechanism in the dissemination of resistance genes among bacterial populations [274]. The interplay among antibiotic exposure, sRNA regulation, and genetic adaptation underscores the critical role of sRNAs in bacterial survival and evolutionary dynamics in response to antimicrobial pressure.

Expanding our understanding of sRNA dynamics reveals new strategies to combat antimicrobial resistance by targeting resistance-conferring sRNAs or disrupting key regulatory networks [275]. Synthetic antisense RNAs, for instance, could be designed to selectively inhibit sRNAs that enhance bacterial survival under antibiotic stress. Similarly, antisense oligonucleotides (ASOs) can be designed to hybridize with sRNAs, preventing their interaction with target mRNAs and thereby inhibiting their regulatory effects. CRISPR-based interventions, particularly CRISPR interference (CRISPRi), offer another promising approach by selectively repressing sRNA expression through catalytically inactive Cas proteins [263]. Additionally, small molecules that disrupt sRNA-mRNA interactions could serve as novel antibiotics, interfering with bacterial adaptation mechanisms. These approaches highlight the growing recognition of sRNAs as viable drug targets, although challenges remain in optimizing their delivery and specificity for clinical applications. Advances in computational tools such as *sRNAdeep* [276] and *TargetRNA3* [277] have also facilitated the accurate prediction and characterization of sRNAs, enabling more precise functional studies. These machine learning-based approaches have significantly improved the identification of bacterial sRNAs and their regulatory targets, providing researchers with powerful tools for elucidating RNA-based regulatory mechanisms. The continued development of high-throughput RNA sequencing techniques and computational models will be instrumental in advancing our understanding of sRNA-mediated gene regulation and its implications for bacterial pathogenesis and antibiotic resistance. As AMR continues to pose a significant global health challenge, further research into sRNA-mediated regulatory mechanisms will be essential for developing novel therapeutic strategies aimed at mitigating the impact of drug-resistant bacterial infections.

## Figures and Tables

**Figure 1 ncrna-11-00036-f001:**
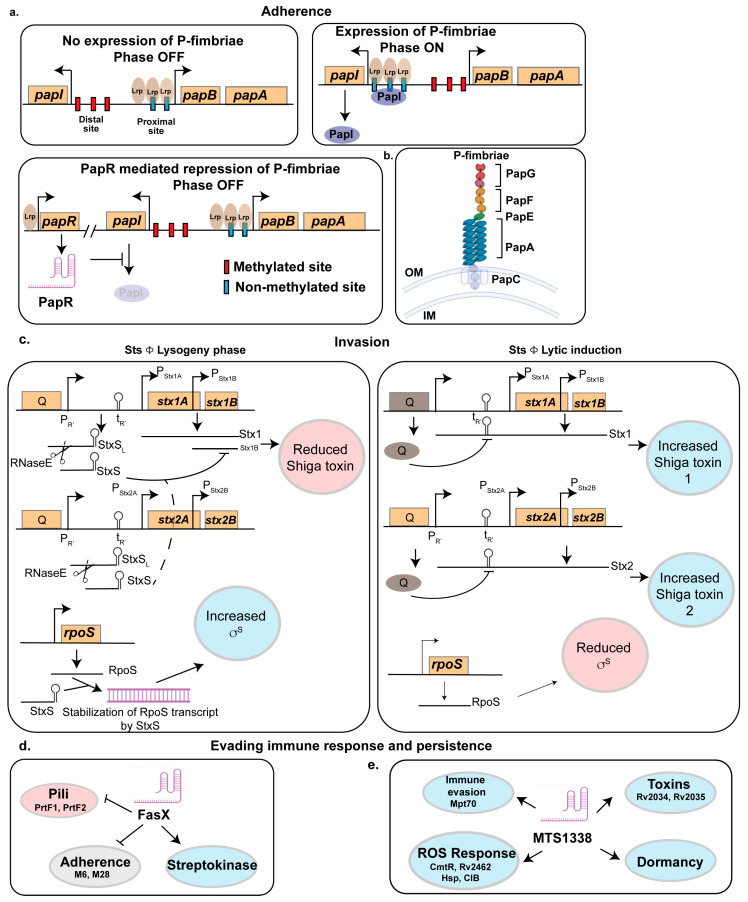
**The role of bacterial sRNAs in virulence**. (**a**) **The regulation of P-pili expression by PapR:** The expression of P-pili in UPEC is controlled by a phase switch that toggles between the OFF and ON states. **Upper left panel:** In the OFF state, the phase switch is regulated by the interaction of Lrp, PapI, PapB, and Dam methylase. The intergenic region between *papB* and *papI* contains two sets of methylation sites: proximal and distal. Binding of Lrp to the proximal site prevents Dam methylation, leading to methylation of the distal site and the suppression of *pap*-operon transcription, thereby preventing P-fimbriae expression. **Upper right panel:** In the ON state, Lrp, in association with PapI, binds to the proximal site, facilitating the activation of the phase switch and allowing transcription of the *pap*-operon, leading to P-fimbriae expression. **Lower left panel:** The small RNA PapR posttranscriptionally represses PapI expression by directly interacting with its transcript, preventing ribosome binding. In the absence of PapI, Lrp cannot activate the phase switch, thereby repressing P-fimbriae expression. A few components of this figure were created using Biorender. (**b**) **The structural components of P-fimbriae:** A schematic representation of the major structural components of P-fimbriae, which facilitate bacterial adherence to host epithelial cells. (**c**) **The role of StxS in Shiga toxin phage regulation: Left panel (Lysogeny):** During lysogeny, the late promoters (PR′) of Stx1 and Stx2 phages are constitutively active, transcribing StxSL, which terminates at the intrinsic terminator tR′. The StxSL transcript is processed by RNase E to generate the regulatory small RNA StxS. StxS enhances the expression of *rpoS*, the stationary phase sigma factor, while repressing *stx1B*, encoding the Shiga toxin 1B subunit, thereby reducing Shiga toxin production. **Right panel (Lytic induction):** During the lytic cycle, the phage-encoded antiterminator Q is expressed, allowing transcriptional readthrough at tR′. The PR′ promoter drives transcription beyond tR′, leading to the expression of Shiga toxin genes. In the absence of StxS, *rpoS* is constitutively expressed. (**d**,**e**) **The role of FasX and MTS1338 in immune evasion and persistence:** Schematic representations of the regulatory roles of FasX and MTS1338 sRNAs in evading the host immune response and promoting bacterial persistence within the host environment.

**Figure 2 ncrna-11-00036-f002:**
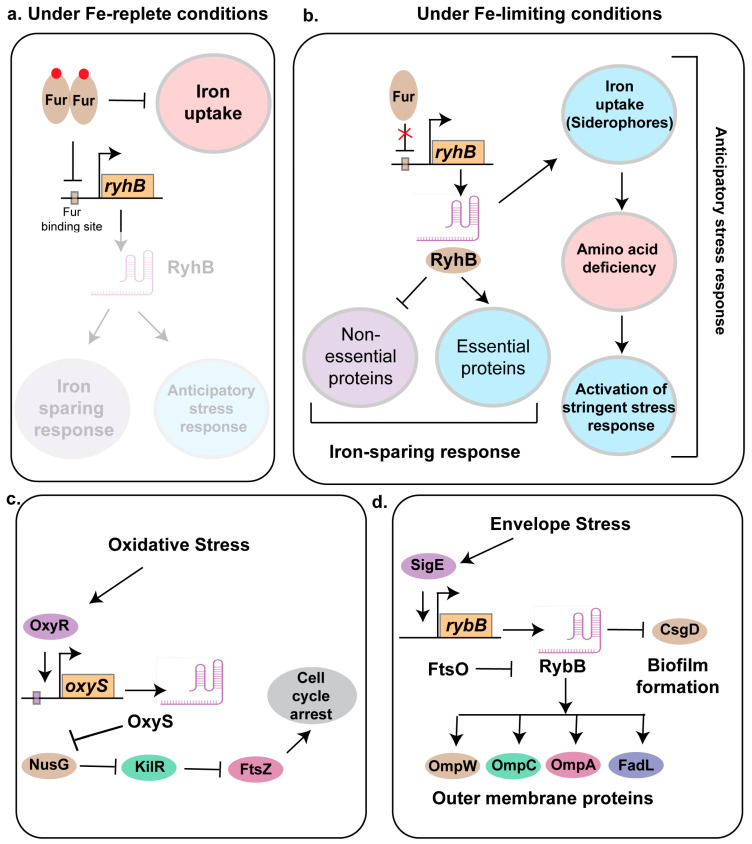
**The role of small RNAs in stress responses**. (**a**) **The regulation of RyhB under iron-replete conditions:** Under Fe-replete conditions, the ferric uptake regulator (Fur) binds to the iron (red circles), becomes active, and represses the transcription of *ryhB* as well as iron uptake systems, preventing unnecessary iron acquisition. A few components of this figure were created using Biorender. (**b**) **The regulation of RyhB under iron-limiting conditions:** Under iron-limiting conditions, Fur becomes inactive, leading to the de-repression of *ryhB* expression. RyhB facilitates iron-sparing responses by selectively repressing the translation of non-essential iron-dependent proteins while allowing the synthesis of essential Fe- and Fe-S cluster-containing proteins. In UPEC strain CFT073, RyhB also contributes to an anticipatory stress response, equipping the pathogen for survival in the nutrient-limited environment of the urinary tract. (**c**) **The role of OxyS in oxidative stress response:** OxyS is induced under oxidative stress conditions by the transcriptional regulator OxyR. OxyS modulates downstream gene expression to mediate DNA damage repair and induce cell cycle arrest, thereby preventing oxidative damage. (**d**) **The role of RybB in envelope stress response:** RybB is activated by the extracytoplasmic stress sigma factor, σ^E^ (SigE/RpoE), under envelope stress conditions. It promotes the synthesis of outer membrane proteins to facilitate membrane repair while simultaneously repressing biofilm formation, thereby balancing cellular adaptation to stress.

**Figure 3 ncrna-11-00036-f003:**
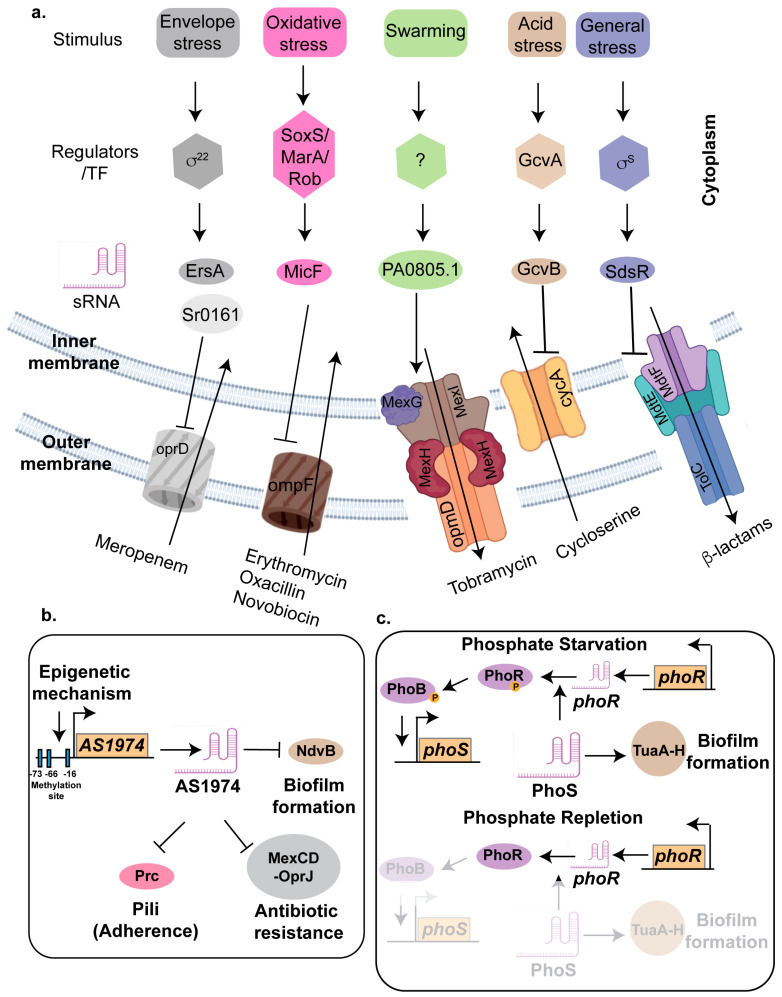
**The mechanisms of bacterial antibiotic resistance.** (**a**) **Small RNA-mediated antibiotic resistance mechanisms:** Upon exposure to environmental stimuli, bacterial pathogens activate transcriptional regulators/factors (TF), which in turn induce the expression of cognate sRNAs. These sRNAs contribute to antibiotic resistance by either upregulating efflux pumps that expel antibiotics from the cell or downregulating outer membrane proteins involved in antibiotic uptake, thereby reducing intracellular antibiotic accumulation. A few components of this figure were created using Biorender. (**b**,**c**) **The role of small RNAs in antibiotic resistance through biofilm formation**. (**b**) **The role of AS1974 in biofilm formation:** The sRNA AS1974 is activated through epigenetic regulation and functions to inhibit antibiotic resistance, biofilm formation, and bacterial adherence to the host. (**c**) **The role of PhoS in biofilm formation:** Under phosphate-limiting conditions, the sensor kinase PhoR undergoes autophosphorylation and subsequently phosphorylates the response regulator PhoB. Activated PhoB induces the expression of *phoS*, an sRNA that serves as a positive regulator of biofilm formation. Additionally, PhoS posttranscriptionally stabilizes *phoR* mRNA, reinforcing a positive feedback loop that sustains the phosphate starvation response. Conversely, under phosphate-replete conditions, PhoR functions as a phosphatase, dephosphorylating PhoB, which in turn prevents *phoS* expression. The absence of PhoS leads to reduced biofilm formation, as indicated by the masked region in the figure.

**Figure 4 ncrna-11-00036-f004:**
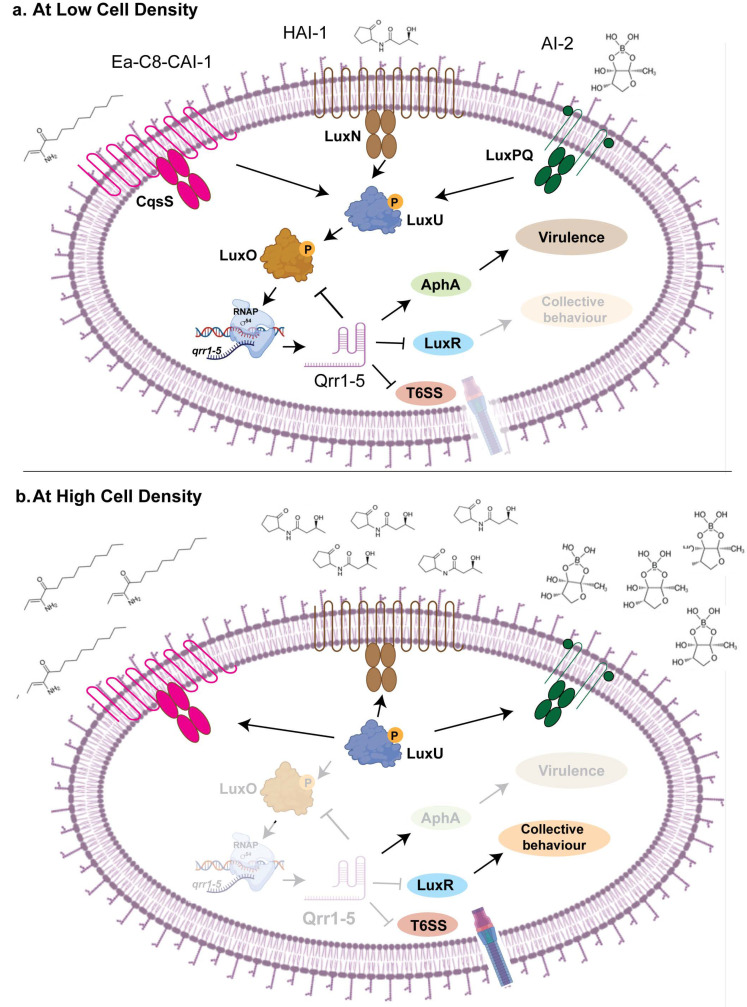
**The role of Qrr sRNAs in quorum sensing regulation in *Vibrio*** spp. ((**a**) **Quorum sensing at low cell density**): At low cell density, the concentration of autoinducers is low, leading to the activation of the sensor kinase, which subsequently phosphorylates downstream regulators. This phosphorylation cascade induces the expression of quorum regulatory RNAs (Qrr sRNAs). Qrr sRNAs function to repress the master quorum sensing regulator LuxR, thereby inhibiting collective behaviors associated with high-density populations. Concurrently, Qrr activates AphA, a key regulator of virulence gene expression, enhancing pathogenicity. Additionally, Qrr sRNAs suppress the type VI secretion system (T6SS), a mechanism involved in interbacterial competition. ((**b**) **Quorum sensing at high cell density**): At high cell density, the accumulation of autoinducers triggers a shift in sensor kinase activity from phosphorylation to phosphatase activity, leading to the degradation of Qrr sRNAs. In the absence of Qrr, LuxR expression is de-repressed, promoting collective behaviors characteristic of quorum sensing, including biofilm formation and luminescence. Additionally, the de-repression of LuxR enhances T6SS activity while simultaneously repressing virulence gene expression, shifting the bacterial population toward cooperative rather than pathogenic behaviors.

## Data Availability

No new data were created or analyzed in this study.

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
