# Peer review of "Tiny but Mighty: Small RNAs—The Micromanagers of Bacterial Survival, Virulence, and Host–Pathogen Interactions"

_ncrna, 2025, doi:10.3390/ncrna11030036_

Round 1
Reviewer 1 Report
Comments and Suggestions for Authors
The submitted manuscript by Dr. Banerjee represents a comprehensive review of the role of small RNAs in bacterial pathogenicity and adaptation. Overall the manuscript is clearly presented although a few basic aspects of review synthesis should be addressed, together with a thorough proof reading to eliminate inconsistent styles of capitalization, italicization and hyphenation.
Specific comments
- There have been multiple reviews on sRNAs so it is important for the author to indicate the need for the current review. How does it differ in scope? Are recent reviews cited? Also, how were the examples selected, and what databases were interrogated?
- L3 Interactions (please use consistent style of capitalization in ALL headers and subleaders through the manuscript)
- L21 governed by sRNAs is essential (the abbreviation was introduced L10)
- L31 The term symbiotic has changed meaning over the decades. It now means prolonged interactions which can include pathogenesis. I would use “mutualistic”
- L65 although many new sRNAs have been identified, some were first identified several decades ago
- L65, L67 and throughout. Please use the abbreviation “sRNA” alone from here on in the ms.
- L72 Please rephrase as it is unclear. It reads as though sRNAs are so named if they encode a small protein, but most examples do not.
- L92 Please delete the period before references
- L102 Please define abbreviation AMR here, then just use AMR on L104
- L143 There are non-sRNA orchestrators also, so I would remove “the”
- L145, L146 (two instances). There may be extra spaces in the text (may be a PDF conversion issue) Also L488? L721?
- L153 Please avoid using etc. as the reader will not know what the author has in their mind regarding the other examples
- L154 and throughout. The genera can be abbreviated after their first mention.
- L156 “and” should not be italicized.
- L156 These serovars should be written capitalized and not italicized. They are not species of Salmonella; also L333, L334, L342, L1499. The correct style is used L767
- L158 Upon adherence, some bacteria deploy
- L165 genus should be italicized
- L175 are the most
- L182 play a crucial
- L190/1. Depending on journal style in vitro, in silico, in vivo should all be italicized (variable italics are used) L428, L429 Correct italics used L545. L1063, L1431, L1464
- L191 settings. UPEC is the primary (UPEC has already been defined L173)
- L192 agent of UTIs (the abbreviation was introduced L183)
- L199 receptors. UPEC strains
- L202 vs L203 P-fimbriae or P fimbriae? Please use one style throughout ms
- L202 associated with UTI pathogenesis
- L207 A recent study….please provide ref for the recent study at the end of the sentence
- L220 a 180-nt sRNA
- L225 There are dozens of examples where it is unclear if the author is referring to the sRNA, the gene or a protein. Genes should all be italicized and not capitalized. Proteins should be capitalized and not italics. In this example, lrp, does not match either style. If the gene deletion is being referred to, it should be italicized. If the protein, then non-italics but capitalized.
- L225 PapR? Italics? Capitalized? please also add a reference at end of L225 and one or more in the half dozen lines of the following paragraph.
- L229 UTI189 probably should not be italicized as it is the strain
- L234 PapI? See pt 28 above. Also L237, L238, L267 (now it is italics)
- L242 400-bp Pap
- L257 P-pili in UPEC is controlled
- L259 papI, papB genes? Proteins? ncRNAs? Please use appropriate style
- Panel C Fig 1 rpoS should be italics if the gene, capitalized if RpoS protein
- L288 are regulated
- L288 It is a 305-nt sRNA
- L290 E. (genus), L292 S. (genus)
- L290 in UPEC strain CFT073
- L301 papC and papC2 styles? (see pt 28)
- L303 murine UTI model
- L313 Please add a reference
- L333 revealed it shares
- L353 P. (genus)
- L355 Clostridiodes difficile (current name)
- L361 S. (genus)
- L381 74-nt RNA
- L394 Please insert space after “gut”
- L404,L409 Agr (when referring to system), agr (italics) for operon
- L409 514-nt sRNA
- L433 E. (genus)
- L444 Bacterial sRNAs can
- L446 cry (italics for genes)
- L448 51-nt sRNA
- L468 The sRNAs discussed in the
- L472 strains
- L473 A period should be directly after “bacteria” The next sentence requires fixing as it splits over two paragraphs
- L497 205-nt sRNA
- L498 It is regulated
- L500 fasX (italics) promoter
- L501 Has FCT been defined?
- L509 and throughout text. Variable hyphenation of posttranscriptional –please unify throughout
- L509/510 prtF1 prtF2 (see pt 28)
- L510 5’ UTRs, specifically (UTR has already been defined so please use abbreviation from that point on)
- L535 mpt70? (pt 8)
- L538 cmtR (pt 28)
- L542 and from this point on, use Mtb abbreviation (introduced L523), L557, L702
- L553 Please add a citation for this factual statement (also L555)
- L560 repertoire of sRNAs to regulate
- L570 peptides plus ROS and RNS produced by phagocytes (abbreviations have already been introduced)
- L580 To combat ROS and RNS, bacteria
- L593 formation of ROS through
- L596 bacterial name (italics)
- L598 with a stress response
- L601 95-nt sRNA that L605 in the TCA
- L606 regulator, Fur [136]. (it had already been described in L5960
- L611 fur (italics? Gene? See pt 28)
- L616 in UPEC strain CFT073
- L621 This is not really unique to one strain; presumably other UPECs use this?
- L623 We found that the absence
- L624 that mimics an Fe-limiting
- L628 distinct from that in commensal
- L632 of an avian
- L634 contributes to
- L636 Persistor formation
- L637 employed by bacteria
- L639 two copies of RyhB
- L644 repressed by Fur
- L654 Please name the gene products to aid the reader
- L658 provides an iron
- Fig 2D Envelope Stress (header) OMPS should be capitalized FadL, OmpW etc
- 2A legend should indicate that red circles are Fe ions
- L667 Responses
- L676 In UPEC strain CFT073
- L684 vs L783 sigma factor or Sigma Factor? I would use lower case example throughout
- L690 is a 100-nt sRNA
- L701 E. (genus)
- L703 infection. It is
- L715 117-nt sRNA in E. coli
- L721 Please define NMR and SAXS
- L731 oxyS (italics?) See pt 28
- L747 9 were linked
- L753 in the stress
- L756 in eukaryotic
- L759 87-nt sRNA L760 posttranscriptionally (or with hyphen) by binding
- L761 to the 5’UTR
- L761 of the rpos (italics?) transcript
- L764 in the acid stress response including the hde
- L765 as well as in pathogenic E. coli
- L770 A GFP-
- L773 Please use either “Furthermore” or “additionally” but not both
- L783 In E. coli
- L787 SigE (protein? Pt 28)
- L789 envelope
- L790 79-nt sRNA
- L791 E. coli
- L798 to cope with envelope stress
- L802 cell division, acting as an RNA sponge (please also indicate what an RNA sponge is, somewhere relevant in the text)
- L809 MicA has not been mentioned yet
- L818 capitalization to match L567 (for example)
- L822 in recent
- L823 has gone
- L823 vs L850 two different styles of hyphenation; please use one style throughout
- L825 and target essential
- L826 thereby killing (or inhibiting) the pathogen. Some of your examples are bacteriostatic (e.g tetracycline) so “killing” can not solely be used here
- L834 DNA replication
- L840 To curb the effects of antibiotics …to develop elaborate…that will be discussed in the following
- L841 treatment with last-line …..and vancomycin of methicillin
- L844 responses to sub-MIC (has MIC been defined yet?)
- L846 genus (italics) serovar (non-italics)
- L849 divided the following
- L855 entry, certain bacteria employ (not all of them use this approach)
- L860 a 205-nt sRNA
- L861 gene; the latter encodes
- L862 In the presence of glycine, transcription
- L863 the acid
- L864 leads to heightened
- L866 spectinomycin is not an aminoglycoside but an aminocyclitol
- L870 cycA (italics? Gene? Pt 28); also L884
- L873 E. coli
- L876 cycloserine is NOT an aminoglycoside
- L882 regions of
- L884 Is DCS-D-cycloserine? It has not been defined
- L886 RpoS (protein? Pt 28)
- L887 SigE? Protein?
- L890 sigE (two instances; pt 28)
- L891 RpoS (here it is a protein)
- L892 approaches
- L893 regulating the type
- Fig 3A define TF in legend. Is sig 22the same as sigE? Only sigE is used in text
- Please check the Fig and txt, In the text, cycloserine enters through CycA, not cirA
- Fig 3B is ndvB a gene or protein (style does not match either), same for mex and prc, tua in 3D
- Fig 3D Phosphate Repletion (to match style of Starvation)
- L920 The 93-nt micF (pt 28) sRNA …. E. coli
- L927 micF (pt 28)
- L934 in the otherwise
- L936 polymyxin (to match capitalization used elsewhere)
- L940 5’ UTR of
- L953 with its 5’ UTR
- :L955 ersA only gets mentioned once
- L959 capitalization of subheaders
- L963 for the polymyxin class
- L965 which changes the charge
- L966 in E.
- L967 modifies LPS with
- L968 by the sRNA MgrR
- L978 of the bacterial
- L984 The emergence of MRSA strains
- L987 within a pathogenicity island
- L988 and is present
- L999 two instances---genes require italics
- L1000 what is “it”? SprX?
- L1008 please add space after mRNA
- L1009 P.
- L1022 please match styles of headers (e.g. sec 4.4)
- L1024 play crucial roles
- L1030 ygiC (italics to match gene ygiB?)
- L1036-1039. Please clearly link references to these statements
- L1049 FQs inhibit gyrase and topo IV, not DNA helicase
- L1050 P. L1050 E.
- L1052, L1053, L1054, L1058, L1061, L1062 (twice), L1064, L1066 emrB? (pt 28)
- L1057 5’ UTR ner the SD sequence
- L1064 deltacsiR (italics if gene)
- L1065 in the presence
- L1067 P.
- L1068 please add a space after “therapy”
- L1075 post-transcriptional (or no hyphen) regulation by sRNAs remains
- L1083 proteins identified. Among these
- L850 vs L1085 multi-drug or multidrug (please use one throughout ms)
- L1099 E.
- L1103 encoding an inner
- L1116/L1117 duplicated part of sentence
- L1133 P.
- L1139 E.
- L1140 resulting in
- L1141 S.
- L1146 V.
- L1152 Although sRNAs play
- L1154 vs L1177 MDR is used with different hyphenation, and one uses it for resistance, the other for resistant.
- L1155 P.
- L1159 should provide new
- L1163 The 127-nt sRNA
- L1169 and prevent their entry
- L1175 and
- L1179 hinting toward
- L1180 akin to methylation
- L1182 strains which are missing in susceptible strains
- L1191 PhoS is a 66-nt sRNA that is
- L1191 phoR (gene; italics)
- L1204 Please link the two paragraphs and avoid one sentence paragraph units
- L1206 reduced EPS production
- L1214 phoS (italics; gene)
- L1217 what is eps here? EPS as in L1206?
- L1217 B. (two instances for genus abbreviation)
- L1223 in a C.
- L1233 capitalization of main headers
- L1241 Gram-positive
- L1243 P. L1243 S.
- L1244 V. L1244 these pathogens use QS
- L1257 P. L1257 S.
- L1261 V. (two instances)
- L1261 is controlled by sRNAs called
- L1293 V.
- L1302 Please define HCD on first use
- L1309 V.
- L1334 In P.
- L1335 las, chl etc (genes? Proteins? Systems?) Pt 28
- L1337 LasI and LasR should be proteins here (not italics)
- L1338, L1339 (pt 28)
- L1343 generating ROS inducing
- L1345 Please add space after [245]. The
- L1345 this operon should be lower case for gene abbreviation (ambBCDE)
- L1348 The sRNA PhrS
- L1365 P.
- L1366 Las and Pqs QS systems
- L1368 Las L1369 Pqs
- L1370 pqs (italics; gene?)
- L1371 sequestering the SD sequence
- L1383 pqsC? Italics? Pt 28
- L1377 PYO (also L1385, L1387, L1391)
- L1384 P.
- L1388 The 100-nt sRNA
- L1388 ami and operon letters should be italicized
- L1391/2 LasI and Rhl? Pt 28 (also L1396)
- L1396 bacterial name should be in italics
- L1404 Pqs QS
- L1411 “pathogen” (no italics) S. aureus (italics) utilizes QS to regulate
- L1413 The Agr system
- L1421 in QS in
- L1422 S.
- L1428 (2 instances), L1429 vs L1430 SprD (italics? Protein? ncRNA? Pt 28)
- L1430 S.
- L1432 near ots SD sequence
- L1437 E. L1437 P. L1437 S.
- L1444 bacterial sRNAs has
- L1446 E.
- L1473 of novel sRNAs that
- L1483 trends in QS research
- L1494 Exposure to sub-MICs of antibiotics
- L1499 SroA (if gene, lower case pt 28)
- L1503 Mtb, MRSA and P. putida (is the latter a pathogen?)
- L1508 rprA (gene? Italics and lower case? Pt 28). Also sdsR L1511.
- L1531 As AMR continues
- L1540 The author declares
- Please check all references. Refs 1 and 2 use different journal styles. See also 21 and 24 (same journal; 2 styles). Please use one style throughout.
I have included many examples above. There may be others in addition. Please thoroughly proof a revised document.
Reviewer 2 Report
Comments and Suggestions for Authors
Overall, this is an exceptionally well-researched and clearly written review about the functions of sRNAs in bacteria related to survival, biofilm formation, and host-pathogen interactions. I have minimal comments, as the review is excellently organized. A broad comment about citation would be to ensure to cite any programs you used to generate figures (e.g., BioRender) if they were not drawn independently.
I believe the content of this review is thorough and an accurate representation of the broad field the author is describing. The citations are largely appropriate and complete.
Minor comments below:
Line 30: Need a citation for the ratio of bacteria to human cells.
Line 40: Mention AMPs disrupt bacterial membranes, but HIV is not a bacterium—perhaps just remove the word “bacterial” in line 38.
Line 55: chemical messengers, collectively called quorum sensors.
Line 71: Typo at beginning of line
Line 129: Need a citation(s) for the processes listed as regulated by quorum sensing
In Figure 1, the majority of the figure is very helpful and well constructed. I suggest putting an “X” ahead of where the rpoS is being transcribed in the “Invasion/ Sts lytic function” section, as it is unclear why more transcription of rpoSwould lead to less sS .
Given the importance of chaperoning these sRNAs, it would be reasonable to include greater mention and citation of Hfq review literature.
Line 683: In E. coli, the envelope stress response protein is commonly referred to as RpoE (not SigE), and therefore you may want to indicate both or use only the Greek letter nomenclature.
Figure 2 and Figure 3 have a typo –should be “Envelope Stress”
Reviewer 3 Report
Comments and Suggestions for Authors
The manuscript provides a comprehensive review of the critical roles of small regulatory RNAs (sRNAs) in bacterial pathogenesis. It highlights how sRNAs contribute to bacterial survival, immune evasion, and antibiotic resistance by regulating gene expression at the post-transcriptional level. The review explores their influence on virulence factor expression, stress response mechanisms, and quorum sensing, emphasizing their potential as targets for novel antimicrobial therapies. By compiling recent research on bacterial sRNAs, the manuscript underscores their significance in host-pathogen interactions and disease progression.
I think the manuscript makes a contribution to microbiology research. It outlines the diverse mechanisms by which sRNAs function, including mRNA degradation, translation repression, and protein sequestration. It includes findings from key pathogens such as Escherichia coli, Salmonella enterica, Vibrio cholerae, Pseudomonas aeruginosa, and Listeria monocytogenes, demonstrating the broad significance of sRNAs in bacterial pathogenesis. The discussion on sRNAs as regulators of antibiotic resistance, including their role in efflux pump regulation and biofilm formation, is particularly valuable given the urgent need for novel antimicrobial strategies. It also provides insights into novel antimicrobial approaches, which could inspire further research into RNA-based drug development.
Some suggestions:
- The review focuses on bacterial regulatory mechanisms but could expand on how sRNAs interact with host immune responses. Suggestion: Include a discussion on how host-derived signals influence sRNA expression and bacterial adaptation during infection.
- While the review mentions bioinformatics and computational predictions, it lacks a detailed discussion on experimental techniques used to identify and characterize bacterial sRNAs. Suggestion: Provide an overview of methodologies such as RNA sequencing (RNA-seq), cross-linking immunoprecipitation (CLIP), and CRISPR-based sRNA functional studies.
- The manuscript briefly discusses the potential of sRNAs as drug targets but does not elaborate on current strategies to inhibit or manipulate bacterial sRNA function. Suggestion: Include examples of antisense oligonucleotides, CRISPR-based interventions, and small molecules designed to disrupt sRNA-mRNA interactions.
- Some sections discuss sRNA functions in specific bacteria, but a comparative analysis highlighting conserved vs. species-specific sRNA mechanisms would strengthen the review. Suggestion: Add a table or figure summarizing key sRNAs across different bacterial pathogens and their conserved vs. unique functions.
- Some statements, particularly those discussing sRNA-mediated
Overall, this manuscript presents a well-documented review of bacterial sRNAs and their roles in host-pathogen interactions. I recommend for revision.
Round 2
Reviewer 2 Report
Comments and Suggestions for Authors
I appreciate the revisions and believe it is ready to be published.